# The *Drosophila* Nab2 RNA binding protein inhibits m⁶A methylation and male-specific splicing of *Sex lethal* transcript in female neuronal tissue

Binta Jalloh[1,2,3†], Carly L Lancaster[1,2,4†], J Christopher Rounds[1,2,3‡], Brianna E Brown[1,2‡], Sara W Leung[1], Ayan Banerjee[1], Derrick J Morton[1,5], Rick S Bienkowski[1,2,3], Milo B Fasken[1], Isaac J Kremsky[1], Matthew Tegowski[6], Kate Meyer[6,7], Anita Corbett[1*], Ken Moberg[2*]

[1]Department of Biology, Emory University, Atlanta, United States; [2]Department of Cell Biology Emory University School of Medicine, Atlanta, United States; [3]Graduate Program in Genetics and Molecular Biology, Emory University, Atlanta, United States; [4]Graduate Program in Biochemistry, Cell and Developmental Biology, Emory University, Atlanta, United States; [5]Emory Institutional Research and Academic Career Development Award (IRACDA), Fellowships in Research and Science Teaching (FIRST) Postdoctoral Fellowship, Atlanta, United States; [6]Department of Biochemistry, Duke University School of Medicine, Durham, United States; [7]Department of Neurobiology, Duke University School of Medicine, Durham, United States

**\*For correspondence:**
acorbe2@emory.edu (AC);
kmoberg@emory.edu (KM)

†These authors contributed
equally to this work
‡These authors also contributed
equally to this work

**Competing interest:** The authors
declare that no competing
interests exist.

**Reviewing Editor:** Douglas L
Black, University of California,
Los Angeles, United States

**Abstract** The *Drosophila* polyadenosine RNA binding protein Nab2, which is orthologous to a human protein lost in a form of inherited intellectual disability, controls adult locomotion, axon projection, dendritic arborization, and memory through a largely undefined set of target RNAs. Here, we show a specific role for Nab2 in regulating splicing of ~150 exons/introns in the head transcriptome and focus on retention of a male-specific exon in the sex determination factor *Sex-lethal* (*Sxl*) that is enriched in female neurons. Previous studies have revealed that this splicing event is regulated in females by N6-methyladenosine (m⁶A) modification by the Mettl3 complex. At a molecular level, Nab2 associates with *Sxl* pre-mRNA in neurons and limits *Sxl* m⁶A methylation at specific sites. In parallel, reducing expression of the Mettl3, Mettl3 complex components, or the m⁶A reader Ythdc1 rescues mutant phenotypes in *Nab2* flies. Overall, these data identify Nab2 as an inhibitor of m⁶A methylation and imply significant overlap between Nab2 and Mettl3 regulated RNAs in neuronal tissue.

## Editor's evaluation

This study provides important new insight into the function of Nab2 protein in regulating Sxl pre-mRNA splicing, a key developmental switch gene in *Drosophila*. By implicating Nab2 in the modulation RNA methylation on the Sxl transcript, the results illuminate the emerging role of methylation in the regulation of splicing. This proposed role for Nab2 in modulating m6A modification has implications for other systems.

## Introduction

RNA binding proteins (RBPs) play important roles in guiding spatiotemporal patterns of gene expression that distinguish different cell types and tissues within organisms. There are an estimated ~1500 RBPs that distribute between the nucleus and cytoplasm (*Gerstberger et al., 2014*), and each has the potential to interact with RNAs to modulate post-transcriptional gene expression. Such regulation is particularly critical in highly specialized cells such as neurons (*Conlon and Manley, 2017*) where regulated alternative splicing of coding regions and 3'UTRs, cleavage/polyadenylation, trafficking, and local translation contribute to precise regulation of gene expression (*Brinegar and Cooper, 2016*). The critical roles of RBPs in neurons are highlighted by many studies that reveal the importance of this class of proteins in brain development and function (*Darnell and Richter, 2012*) and by the prevalence of human neurological diseases linked to mutations in genes encoding RBPs (*Brinegar and Cooper, 2016*). Many of these RBPs are ubiquitously expressed and play multiple roles in post-transcriptional regulation. Thus, defining the key neuronal functions of these proteins is critical to understanding both their fundamental roles and the links to disease.

Among the RBPs linked to human diseases are a group of proteins that bind with high affinity to polyadenosine RNAs, which are termed poly(A) RNA binding proteins or Pabs (*Kelly et al., 2010*). Functional studies of classical nuclear and cytoplasmic Pabs, which utilize RNA recognition motifs (RRMs) to recognize RNA, have uncovered diverse roles for these proteins in modulating mRNA stability, alternative cleavage, and polyadenylation and translation (*Smith et al., 2014*). A second, less well-studied, group of Pabs uses zinc-finger (ZnF) domains to bind target RNAs. Among these is the zinc finger Cys-Cys-Cys-His-type containing 14 (ZC3H14) protein, which binds with high affinity to poly(A) RNAs via a set of C-terminal tandem Cys-Cys-Cys-His type ZnF domains (*Leung et al., 2009*). ZC3H14 is broadly expressed in many tissues and cell types but mutations in the human *ZC3H14* gene are associated with a heritable form of intellectual disability (*Pak et al., 2011*), implying an important requirement for this protein in the central nervous system.

ZC3H14 has well-conserved homologs in eukaryotes, including *Saccharomyces cerevisiae* Nuclear poly(A)-binding protein 2 (Nab2), *Drosophila melanogaster* Nab2, *Caenorhabditis elegans* SUT-2, and murine ZC3H14 (*Fasken et al., 2019*). Zygotic loss of *ZC3H14* in mice and *Drosophila* impairs neuronal function (*Pak et al., 2011*; *Roxas et al., 2017*), while neuron-specific depletion of *Drosophila* Nab2 is sufficient to replicate these effects (*Pak et al., 2011*). Reciprocally, expression of human ZC3H14 in Nab2-deficient neurons rescues this defect, demonstrating a high degree of functional conservation between human ZC3H14 and *Drosophila* Nab2 (*Kelly et al., 2014*). Collectively, these data focus attention on what are critical, but poorly understood, molecular roles for ZC3H14/Nab2 proteins in neurons.

Neuronal ZC3H14/Nab2 can be divided into two pools: a nuclear pool that accounts for the majority of ZC3H14/Nab2 in the cell, and a small cytoplasmic pool of protein detected in mRNA ribonucleoprotein particles (mRNPs) of axons and dendrites (*Leung et al., 2009*; *Roxas et al., 2017*; *Bienkowski et al., 2017*). Depletion of both pools in *Drosophila* neurons causes defects in axon projection within the brain mushroom bodies (MBs) (*Kelly et al., 2016*), a pair of neuropil structures involved in olfactory learning and memory (*Armstrong et al., 1998*; *Heisenberg, 2003*), and excess branching of dendrites on peripheral sensory neurons (*Corgiat et al., 2022*). The Nab2 requirement in MBs is linked to a physical association between Nab2 and the *Drosophila* Fragile-X mental retardation protein homolog (*Wan et al., 2000*) in the neuronal cytoplasm and translational repression of shared Nab2-Fmr1 target RNAs (*Bienkowski et al., 2017*). Genetic data indicate that Nab2 limits dendritic branching through effects on the cytoplasmic planar cell polarity pathway (*Corgiat et al., 2022*). Despite these insights into a cytoplasmic functions of Nab2, a molecular role of the abundant pool of Nab2 protein in neuronal nuclei remains undefined.

Here, we employ a broad and an unbiased RNA sequencing approach to identify transcriptome-wide changes in the heads of *Nab2* loss-of-function mutant flies. While the steady-state levels of most transcripts are not significantly changed, we find a striking effect on splicing of a subset of neuronal RNA transcripts. We focus our analysis on a well-characterized sex-specific alternative splicing event in the *Sex-lethal* (*Sxl*) transcript (*Salz and Erickson, 2010*; *Förch and Valcárcel, 2003*; *Haussmann et al., 2016*). Results reveal that *Nab2* plays a novel role in regulating the alternative splicing of *Sxl* in a sex-specific manner. Recent work has revealed a role for m⁶A RNA methylation by the enzyme Mettl3 in modulating this splicing event (*Lence et al., 2016*; *Kan et al., 2017*). Similar to *Mettl3*, the

requirement for *Nab2* in alternative splicing of *Sxl* is only essential for neuronally enriched tissues. Genetic and biochemical experiments support a functional link between m⁶A methylation and Nab2 in which Nab2 limits m⁶A on target RNAs. These results demonstrate the role for *Drosophila* Nab2 in RNA alternative splicing as well as RNA methylation and sex determination in neurons.

## Results

### Nab2 loss affects levels and processing of a subset of RNAs in the transcriptome of the *Drosophila* head

To assess the role of Nab2 in regulating the central nervous system transcriptome, a high-throughput RNA sequencing (RNA-Seq) analysis was carried out in triplicate on *Nab2* null mutant heads (*Nab2ex3* imprecise excision of *EP3716*) (*Pak et al., 2011*) and isogenic control heads (*Nab2pex41* precise excision of *EP3716*). To capture any sex-specific differences, heads were collected from both male and female flies of each geno-type. Briefly, total RNA from 1-day-old adults was rRNA-depleted and used to generate stranded cDNA libraries that were sequenced (150 cycles) on a NextSeq 500 High Output Flow Cell. This generated a total of approximately 1.1 billion 75 base-pair (bp) paired-end reads (91 million/sample) that were mapped onto the Dmel6.17 release of the *Drosophila* genome using RNA STAR (*Dobin et al., 2013*). Read annotation and per-gene tabulation was conducted with feature-Counts (*Liao et al., 2014*), and differential expression analysis was performed with DESeq2 (*Love et al., 2014*).

RNA sequencing reads across the *Nab2* gene are almost completely eliminated in *Nab2ex3* mutants, confirming the genetic background and integrity of this analysis pipeline (*Figure 1—figure supplement 1*). Principal component analysis (PCA) performed with DESeq2 output data confirms that the 12 RNA-seq datasets distribute into four clusters that diverge significantly from one another based on genotype (*Nab2ex3* vs. *Nab2pex41* control; PC1 58% variance) and sex (male vs. female; PC2 26% variance) (*Figure 1A*). The DESeq2 analysis detects 3799 and 1545 genes in females and males, respectively, that exhibit statistically significant differences in RNA abundance between *Nab2ex3* and control (BH-adjusted p-value/false discovery rate [FDR] < 0.05) (*Supplementary file 1*). Comparison of fold-changes (*Nab2ex3* vs. control) among these significantly different RNAs reveals a high degree of correlation in female vs. male samples (R = 0.79), particularly among RNAs whose levels are most elevated upon Nab2 loss (*Figure 1B*). Applying a twofold change cutoff (|log₂[fold-change]| ≥ 1) trims these sets to 453 significantly changed RNAs in females (294 'up,' 159 'down') and 305 significantly changed RNAs in males (150 'up,' 155 'down') (*Figure 1C*), which merge into a combined set of 570 significantly affected RNAs that trend similarly in heatmap analysis of mutant vs. control samples (*Figure 1D*). A majority of the 453 affected 'female' RNAs are mRNAs (439) and the remaining are snoRNAs (8), snRNAs (1), pre-rRNAs (1), and tRNAs (4)

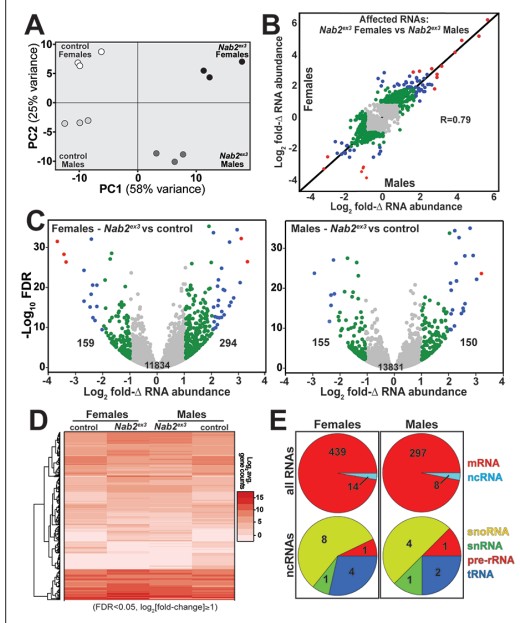

**Figure 1.** RNA sequencing detects effects of Nab2 loss on the head transcriptome. (**A**) Principal component analysis (PCA) of RNA-seq data from three biological replicates of control and *Nab2* mutant (*Nab2ex3*) male and female heads. (**B**) Correlation scatter plot of log₂ fold change (Δ) in abundance of affected RNAs in males and females (log₂ average gene counts: gray < 1, 1 ≤ green < 2, 2 ≤ blue < 3, red ≥ 3). (**C**) Volcano plots of fold-Δ in abundance vs. false discovery rate (FDR -log₁₀) of affected RNAs in *Nab2ex3* females and males (dot plot color coding as in **B**). Elevated (1), reduced (–1), and total RNAs are indicated. (**D**) Heatmap comparison of significantly changed gene counts (FDR < 0.05;|log₂ fold-Δ| 1) in *Nab2ex3* females and males vs. sex-matched controls. (**E**) Pie chart distribution of RNA classes among significantly affected RNAs detected in (**C**) and (**D**).

The online version of this article includes the following figure supplement(s) for figure 1:

**Figure supplement 1.** RNA sequencing reads across the *Nab2* locus.

(*Figure 1E*). A similar distribution occurs in male heads: a majority of the affected RNAs are mRNAs (297) and the remainder are snoRNAs (4), snRNAs (1), pre-rRNAs (1), and tRNAs (2) (*Figure 1E*). Overall, the number of significantly changed RNAs ($|\log_2[\text{fold-change}]| \geq 1$ and FDR < 0.05) in *Nab2^ex3* female and male heads represents a small fraction of RNAs detected in heads (2.2 and 3.7% in males and females, respectively), suggesting that Nab2 normally contributes to RNA-specific regulatory mechanisms in *Drosophila* head tissue.

## Nab2 loss alters levels of transcripts linked to mRNA processing

To identify functional groups within Nab2-regulated RNAs, Gene Set Enrichment Analysis (GSEA) (*Mootha et al., 2003*; *Subramanian et al., 2005*) was performed with the goal of defining enriched Gene Ontology (GO) terms (*Ashburner et al., 2000*; *The Gene Ontology Consortium, 2019*) among the significantly changed female and male RNAs identified by DESeq2. This filtering uncovers significant enrichment (p<0.05) for 'RNA splicing' GO (GO:0008380) within the upregulated group of RNAs in both sexes (*Figure 2A*). In *Nab2^ex3* females, 32 of 155 genes annotated under this term are present among upregulated RNAs; whereas in males, 75 of 159 genes annotated under this term are present among upregulated RNAs (*Figure 2A*). This enrichment for upregulated splicing-related factors indicates that Nab2 loss could shift splicing patterns in the adult head. Consistent with this hypothesis, mixture of isoforms (MISO) analysis (*Katz et al., 2010*) of annotated alternative splicing events confirms that Nab2 loss significantly alters splicing patterns within a small number of transcripts in female (48) and male (50) heads (*Supplementary file 2*) that fall into a variety of GO terms (*Figure 2—figure supplement 1*). These MISO-called alternative splicing events include 5′ and 3′ alternative-splice site usage, intron retention events, and previously annotated exon skipping events, some of which are detected in the same transcripts (*Figure 2B*).

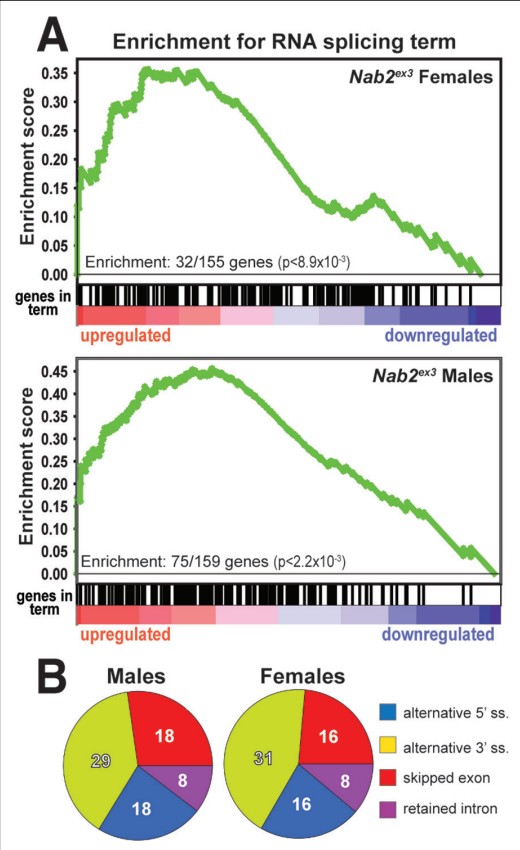

**Figure 2.** Significantly up-/downregulated RNAs in *Nab2^ex3* heads are enriched for predicated splicing factors. (**A**) Gene Set Enrichment Analysis (GSEA) detects enrichment for the 'RNA splicing' Gene Ontology (GO) term in up- and downregulated gene sets in both female (top) and male (bottom) *Nab2^ex3* datasets. Gene enrichments are indicated with corresponding p-values. (**B**) Pie chart illustrating the distribution of previously annotated alternative splicing RNA splicing events that are significantly altered in *Nab2^ex3* mutant female and male heads (ss = splice site).

The online version of this article includes the following figure supplement(s) for figure 2:

**Figure supplement 1.** Gene Ontology (GO) term enrichment among Nab2-regulated alternative splicing events.

To test whether Nab2 loss results in unannotated or aberrant splicing events among head RNAs, DEXSeq analysis (*Anders et al., 2012*) was performed to scan for differential abundance of individual exons relative to other exons within the same transcript. This analysis detects 151 affected RNAs in *Nab2^ex3* females and 114 in *Nab2^ex3* males (*Table 1*), with many top-ranked transcripts encoding factors with roles in behavior, neurodevelopment, and/or neural function (*Supplementary file 3*). Reanalysis with a lower significance threshold yielded additional transcripts that show evidence of altered post-transcriptional processing in *Nab2* mutant heads but did not alter the group of RNAs identified as most significantly affected by Nab2 loss. Among the 151 most affected RNAs, the most statistically significant exon usage change in either sex is female-specific inclusion of exon 3 in the *Sex lethal* (*Sxl*) mRNA (p=3.08 × 10^−235). This effect on *Sxl* mRNA in *Nab2^ex3* females is followed in rank

**Table 1.** Alternative exon use (DexSeq) in *Nab2*[ex3] head transcriptomes.

| | Females | Males |
|---|---|---|
| No. of alternatively used exons * | 151 | 114 |
| * |(exon usage fold change)| >~1.75 BH Adj. p<0.05 | | |

| Top mis-spliced transcripts | | |
|---|---|---|
| **Females** | foldΔ exon usage | BH adj. p-value |
| *Sex lethal (Sxl)* | 2.86 | $3.08 \times 10^{-235}$ |
| *CG13124* | 2.45 | $1.09 \times 10^{-81}$ |
| *Ih channel* | 2.29 | $3.28 \times 10^{63}$ |
| *Ace* | 1.81 | $1.02 \times 10^{-59}$ |
| **Males** | foldΔ exon usage | BH adj. p-value |
| *Ace* | 2.02 | $1.88 \times 10^{-169}$ |
| *Pkc53E* | 1.74 | $6.12 \times 10^{-102}$ |
| *plx* | 2.12 | $9.03 \times 10^{-67}$ |
| *Pkn* | 1.84 | $9.04 \times 10^{-67}$ |
| *Bacc* | 2.31 | $1.11 \times 10^{-64}$ |

The online version of this article includes the following source data for table 1:

**Source data 1.** Files (.bed format, openable as .excel tables) of splicing defects detected by lower threshold analysis than in *Table 1*.

order of statistical significance by enhanced inclusion of exons 1 and 2 of the MIF4GD homolog transcript *CG13124*, exons 1 and 2 of the voltage-gated ion channel transcript $I_h$ *channel* ($I_h$), and exon 1 of the synaptic enzyme transcript *Acetylcholine esterase* (*Ace*). In *Nab2*[ex3] males, the top four events are enhanced inclusion of exon 1 of the *Ace* transcript, exon 1 of the *Protein kinase C at 53E* (*Pkc53E*) transcript, exons 1 and 2 of the Rab GTPase *pollux* (*plx*) transcript, and exons 1 and 2 of *Protein kinase N* (*Pkn*) transcript. In a number of cases, identical exons are affected in both *Nab2*[ex3] sexes and accompanied by retention of the intervening intron (e.g., see *CG13124* and $I_h$ traces in *Figure 3—figure supplement 1*). The robust increase in *Sxl* exon 3 in *Nab2*[ex3] females is noteworthy both for the central role that differential inclusion of *Sxl* exon 3 plays in *Drosophila* sex determination (*Harrison, 2007*) and because DEXSeq did not detect changes in exon 3 inclusion or abundance in *Nab2*[ex3] males. In light of this sex-specific effect of Nab2 loss on alternative splicing of *Sxl* exon 3, subsequent analyses focused on the role of Nab2 in *Sxl* mRNA splicing in female heads.

### *Nab2*[ex3] females exhibit masculinized *Sxl* splicing in neuron-enriched tissues

The Sxl protein is a female-specific, U-rich RNA binding protein that is best defined for its role acting through the *tra-dsx* and *msl-2* pathways to promote female somatic and germline identity (*Penalva and Sánchez, 2003*; *Gawande et al., 2006*). *Sxl* pre-mRNA is expressed in both males and females, but alternative splicing regulated by m⁶A RNA methylation and several RBPs leads to female-specific skipping of exon 3 during splicing (*Haussmann et al., 2016*; *Lence et al., 2016*; *Sakamoto et al., 1992*). Because exon 3 includes an in-frame translation 'stop' codon, full-length Sxl protein is only made and active in female cells (*Bell et al., 1988*). The inclusion of *Sxl* exon 3 in *Nab2*[ex3] mutants would thus implicate Nab2 as a novel component of molecular machinery that controls *Sxl* pre-mRNA splicing.

Visualizing *Sxl* RNA-Seq reads with Integrative Genomics Viewer (IGV) (*Robinson et al., 2017*) confirms a large increase in exon 3 reads in *Nab2*[ex3] females (*Nab2*[ex3] F) relative to control females (control F), and also reveals retention of ~500 bases of intron 3 sequence in *Nab2*[ex3] females (*Figure 3A*). Normal splicing patterns are detected across all other *Sxl* intron-exon junctions in both

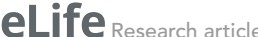

**Figure 3.** *Sxl* alternative splicing and protein levels are disrupted in *Nab2ex3* female heads. (**A**) Top panel: normal *Sxl* alternative splicing patterns across exon 2-4 and exon 8-10 regions in female (F) and male (M). Bottom panel: corresponding sequencing reads across the *Sxl* locus in the indicated sexes and genotypes. Dotted lines and boxed insets highlight exon 3 and exon 9 reads. (**B**) RT-PCR analysis of *Sxl* mRNA in control, *Nab2ex3* and *Mettl3null* female (F) and male (M) heads. Exon 2-3-4 and exon 2-4 bands indicated. Arrowhead denotes exon 2-3-intron-4 product noted in text. Asterisk is

*Figure 3 continued on next page*

*Figure 3 continued*

nonspecific product. (**C**) RT-qPCR analysis of *Sxl* transcripts in adult female, control *Nab2*^ex3^, and *Mettl3*^null^ heads using the indicated primer sets. Asterisk indicates results that are statistically significant at p-value<0.05. (**D**) Immunoblot of protein samples from control, *Nab2*^ex3^, and *Mettl3*^null^ female heads. Antibody against female-specific Sxl protein isoform was used to detect Sxl in each sample. Lamin serves as a loading control. Molecular weights are given in kDa and indicated to the left. (**E**) Quantification of Sxl protein levels in (**D**) using ImageLab software. Protein levels are normalized to control, with the value for control set to 1.0. Asterisk indicates results that are statistically significant at p-value<0.05. (**F**) RT-PCR analysis of *Sxl* mRNA in adult female control and *Nab2*^ex3^ tissues with exon 2-3-4 and 2-4 bands indicated. (**G**) RNA binding protein (RBP) motif enrichment analysis detects predicted Sxl binding sites as the most frequent motif among Nab2-regulated splicing events in female heads. Other enriched motifs are similar between male and female heads. Regions used for motif analysis (retained introns, and alternative 5' or 3' splice sites plus flanking sequence) are described in the text and illustrated in the schematic to the right.

The online version of this article includes the following source data and figure supplement(s) for figure 3:

**Source data 1.** Original agarose gel showing RT-PCR analysis of *Sxl* mRNA in the relevant genotypes in *Figure 3B*.

**Source data 2.** Original western blot of head lysates labeled in *Figure 3D* using anti-Sxl antibody.

**Source data 3.** Original western blot of head lysates labeled in *Figure 3D* using anti-Lamin antibody.

**Source data 4.** Raw agarose gel showing RT-PCR analysis of *Sxl* mRNA in the relevant genotypes and tissues shown in *Figure 3F*.

**Figure supplement 1.** RNA sequencing reads across the *CG13124* and *I*$_h$ *channel* loci.

**Figure supplement 2.** RNA sequencing reads across the *tra* and *dsx* loci.

genotypes of males and females, including female-specific exon 9 inclusion. Quantification of reads across the entire *Sxl* locus detects an ~1.5-fold increase in the overall abundance of the *Sxl* mRNA in *Nab2*^ex3^ females compared to control females. Parallel reverse transcription polymerase chain reaction (RT-PCR) on fly heads using *Sxl* primers that detect exon 2-exon 4 (control females) and exon 2-exon 3-exon 4 (control males) confirms the presence of the mis-spliced exon 2-exon 3-exon 4 mRNA transcript in *Nab2*^ex3^ females (*Figure 3B*). The exon 2-exon 3-exon 4 mRNA transcript appears to be more abundant in *Nab2*^ex3^ female heads than in female heads lacking *Mettl3*, which encodes the catalytic component of the m⁶A methyltransferase complex that promotes exon 3 skipping in nervous system tissue (*Haussmann et al., 2016*; *Lence et al., 2016*; *Kan et al., 2017*). RT-PCR also reveals an ~1 kb band in *Nab2*^ex3^ females (arrowhead, *Figure 3B*) that sequencing identifies as aberrantly spliced transcript that incorporates 503 bases of intron 3 leading up to a cryptic 5' splice site (i.e., exon 2-exon 3-intron 3^503^-exon 4), which matches the *Sxl* intron 3 sequencing reads observed in IGV (see *Figure 3A*).

qRT-PCR confirms a statistically significant increase in the inclusion of the male-specific exon 3 in females with a concomitant decrease in the level of correctly spliced (exon 2-exon 4) transcript in both *Nab2*^ex3^ and *Mettl3*^null^ female heads (*Figure 3C*). Because *Sxl* exon 3 includes an in-frame translation 'stop' codon, we tested whether full-length Sxl protein levels decrease in *Nab2*^ex3^ female heads. Indeed, immunoblotting analysis reveals reduced levels of Sxl protein in *Nab2*^ex3^ female heads compared to control or *Mettl3* null heads (*Figure 3D and E*). Together, these data implicate Nab2 in post-transcriptional regulation of *Sxl* splicing and control of Sxl protein levels within female heads.

As all the analysis carried out thus far employed heads as source material, we tested whether Nab2-dependent splicing changes were also detected in other tissues. Significantly, RT-PCR analysis of *Sxl* mRNA in dissected control and *Nab2*^ex3^ females detects exon 3 retention in *Nab2*^ex3^ samples prepared from the thorax, but little to no retention in the abdomen and ovary (*Figure 3F*). This result implies that Nab2 is only necessary to direct *Sxl* exon 3 exclusion in specific tissues or cell types such as neurons, which are enriched in the head (brain) and thorax (ventral nerve cord). In sum, these data reveal a tissue-specific role for Nab2 in blocking *Sxl* exon 3 inclusion in females and regulating 5'-splice site utilization at the exon 3-exon 4 junction.

As Sxl is itself an RBP with roles in alternative splicing (*Penalva and Sánchez, 2003*; *Bell et al., 1988*), we performed a bioinformatic scan for RBP motifs enriched in proximity to the Nab2-dependent alternative splicing events identified by MISO analysis (see *Figure 2B*). Input sequences were composed of retained introns plus 25 nt extending into each flanking exon, and alternative splice sites with 25 nt of exon plus 1 kb of adjacent intron (see schematic, *Figure 3G*). This unbiased scan detected candidate Sxl binding sites as the single most abundant RBP motif within the Nab2-regulated MISO events in females (*Figure 3G*). Notably, Sxl motifs were not detected as enriched in the male *Nab2*^ex3^ MISO dataset, which otherwise strongly resembles the remaining group of female-enriched RBP motifs (e.g.,

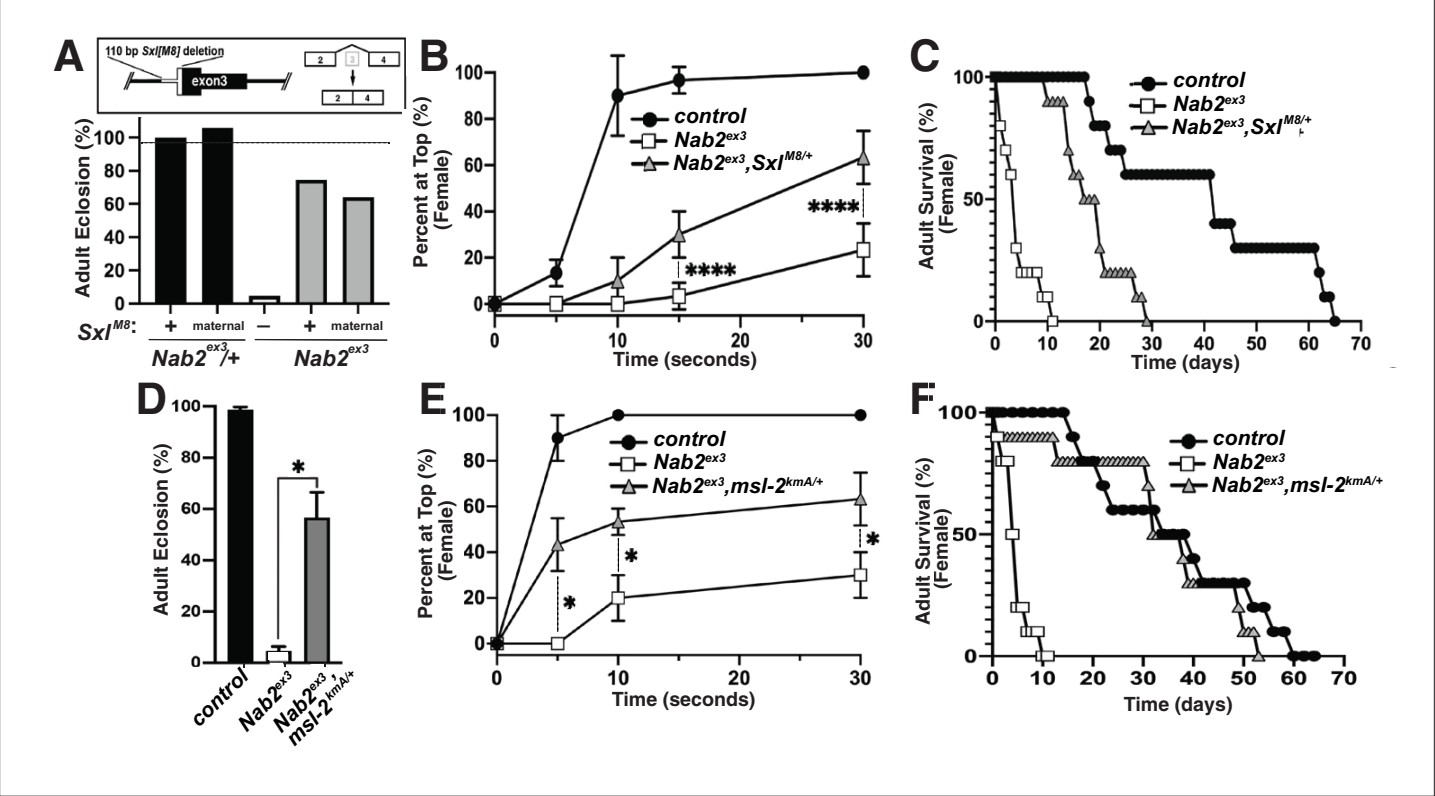

**Figure 4.** Alleles of *Sxl^M8* or the Dosage Compensation Complex (DCC) component *male-specific lethal-2* (*msl-2*) rescues *Nab2* phenotypes. (**A**) A single copy of the *Sxl^M8* allele, which harbors a 110 bp deletion that causes constitutive exon 2-4 splicing, partially suppresses lethality of *Nab2^ex3*, both zygotically and maternally (calculated as #observed/#expected). (**B, C**) *Sxl^M8* dominantly (i.e., *M8/+*) suppresses previously defined (**B**) locomotion (as assessed by negative-geotaxis) and (**C**) lifespan defects in age-matched *Nab2^ex3* females. (**D**) Percent of control, *Nab2^ex3*, or *msl-2^kmA/+;;Nab2^ex3* (*msl-2* is on the X chromosome) that eclose as viable adults (calculated as #observed/#expected). (**E, F**) *msl-2^kmA* dominantly (i.e., *kmA/+*) suppresses previously defined (**E**) locomotion (as assessed by negative-geotaxis) and (**F**) lifespan defects in age-matched *Nab2^ex3* females. Significance values are indicated (*$p<0.05$, ****$p<0.0001$).

The online version of this article includes the following figure supplement(s) for figure 4:

**Figure supplement 1.** Additional genetic interactions between *Nab2^ex3 msl-2*, *roX1*, and *mle*.

the hnRNPL homolog *smooth* [*sm*], *RNA binding protein-9* [*Rbp9*], the U1-SNRNPA homolog *sans fille* [*snf*], and the U2-SNRNP component [*U2AF50*]). The female-specific enrichment for Sxl binding sites raises the possibility that Nab2 regulates a portion of the alternative splicing events indirectly via control of a Sxl-regulated splicing program, or that Sxl and Nab2 proteins target common splicing events. Intriguingly, the Sxl target *transformer* (*tra*) and the Tra target *double-sex* (*dsx*) (*Sánchez et al., 2001*; *Horabin and Schedl, 1993*) were not recovered in the *Nab2^ex3* MISO or DESeq2 datasets, and IGV reads show little evidence of altered structure of *tra* and *dsx* RNAs as compared to *Nab2^pex41* controls (*Figure 3—figure supplement 2*). Together, these data suggest that Sxl may not control the *tra-dsx* pathway in the adult head, or that *tra* and *dsx* splicing are only altered in a subset of *Nab2^ex3* head cells and thus not detectable by bulk RNA-Seq analysis.

## The dosage compensation complex contributes to phenotypes in *Nab2^ex3* mutant females

The decrease in Sxl protein in *Nab2^ex3* female heads suggests that aberrant inclusion of *Sxl* exon 3 could contribute to *Nab2^ex3* phenotypes by reducing Sxl activity. To test this hypothesis, the constitutively female-spliced *Sxl^M8* allele (*Barbash and Cline, 1995*) was placed as a heterozygote into the background of *Nab2^ex3* animals. *Sxl^M8* contains a 110 bp deletion spanning the 3'-end of intron 2 and 5'-end of exon 3, and consequently undergoes constitutive splicing to the feminized exon 2-exon 4 variant regardless of sex (*Figure 4A*, top panel). Consistent with the original report describing *Sxl^M8*

(*Barbash and Cline, 1995*), the allele is male-lethal in both control and *Nab2ex3* backgrounds. However, heterozygosity for *SxlM8* produces strong rescue of *Nab2ex3* mutant female viability from ~4% to 71% (*SxlM8/+;;Nab2ex3*) (*Figure 4A*). Female *Nab2ex3* siblings that did not inherit the *SxlM8* allele also exhibit elevated viability (64%), perhaps due to maternal loading of *Sxl* mRNA. *SxlM8/+;;Nab2ex3* females also show improved locomotion in a negative geotaxis assay (*Figure 4B*) and lengthened lifespan (*Figure 4C*) relative to *Nab2ex3* females. This female-specific rescue of *Nab2ex3* by *SxlM8* indicates that partial restoration of Sxl expression can compensate for Nab2 loss.

The absence of any effect on *tra* or *dsx* transcripts upon loss of Nab2 (see *Figure 3—figure supplement 2*) prompted us to analyze the other major role of Sxl, which is to bind to the *male-specific lethal-2* (*msl-2*) mRNA and inhibit its translation in female somatic and germline tissues (*Lucchesi and Kuroda, 2015*; *Keller and Akhtar, 2015*). As a result, Msl-2 protein is only expressed in male cells, where it promotes assembly of a chromatin modifying complex termed the Dosage Compensation Complex (DCC; composed of Msl-1, Msl-2, Msl-3, Mof, Mle, and *roX1* and *roX2* non-coding RNAs), which is recruited to the male X chromosome to equalize X-linked gene expression between males and females (*Lucchesi and Kuroda, 2015*; *Keller and Akhtar, 2015*). A number of DCC components are expressed at high levels in the adult nervous system (*Amrein and Axel, 1997*), which correlates with the tissue-restricted link between Nab2 and *Sxl* splicing (as in *Figure 3F*). As a functional test of interactions between Nab2 and the DCC pathway, the *msl-2kmA* (*msl-2killer of males-A*) male lethal EMS allele (*Bevan et al., 1993*) was tested for dominant effects on *Nab2ex3* female phenotypes. A single copy of *msl-2kmA* significantly rescues defects in viability (*Figure 4D*), locomotion (*Figure 4E*), and lifespan (*Figure 4F*) in *Nab2ex3* females. Furthermore, a second *msl-2* mutant allele over a deficiency that uncovers the *msl-2* locus (*msl-l227/Exel7016*) (*Zhou et al., 1995*), as well as *roX1* and *mle* loss-of-function alleles, rescue *Nab2ex3* mutant phenotypes (*Figure 4—figure supplement 1*). Given that Msl-2 is not normally active in adult female tissues (*Amrein and Axel, 1997*; *Meller et al., 1997*) and its forced expression reduces female viability (*Kelley et al., 1995*), rescue of *Nab2ex3* females by these *msl-2*, *mle*, and *roX1* alleles provides evidence that the DCC pathway is inappropriately activated upon Nab2 loss. Of note, the *msl-2*, *mle*, and *roX1* RNAs appear similar in IGV reads from both control and *Nab2ex3* adult heads (see *Figure 3—figure supplement 2*), suggesting that genetic interactions between these loci are not through direct effects of Nab2 loss on post-transcriptional processing of these RNAs in a large fraction of cells.

## Loss of the Mettl3 m⁶A methyltransferase rescues *Nab2ex3* phenotypes

Genetic interactions between *Nab2*, *Sxl*, and *msl-2* alleles are consistent with a role for Nab2 protein in regulating sex-specific splicing of *Sxl* exon 3. One mechanism that promotes exon 3 exclusion in females is based on N⁶-methylation of adenosines (m⁶A) in *Sxl* pre-mRNA by the Methyltransferase like-3 (Mettl3)-containing methyltransferase complex (*Haussmann et al., 2016*; *Lence et al., 2016*). Inactivating mutations in components of this m⁶A 'writer' complex masculinize the pattern of exon 3 splicing in female flies in a manner similar to *Nab2ex3* and molecular studies indicate that the Mettl3 complex promotes exon 3 exclusion in females by depositing m⁶A within *Sxl* exon 3 and flanking introns (*Haussmann et al., 2016*; *Lence et al., 2016*; *Kan et al., 2017*; *Kan et al., 2021*).

To assess Nab2-Mettl3 functional interactions, the *Mettl3null* allele (formerly known as *Ime4null*) (*Lence et al., 2016*) was carefully recombined with *Nab2ex3* (the loci are less than 1 cM apart; *Figure 5—figure supplement 1*). Multiple recombinant *Nab2ex3,Mettl3null* chromosomes were found to be lethal at pre-larval stages but semi-viable over the *Nab2ex3* chromosome; we therefore analyzed phenotypes in *Nab2ex3,Mettl3null/+* mutant females. Consistent with prior work (*Haussmann et al., 2016*; *Lence et al., 2016*; *Kan et al., 2017*), homozygosity for the *Mettl3null* allele reduces adult viability (*Figure 5A*), decreases locomotion in a negative geotaxis assay (*Figure 5B*), and shortens lifespan (*Figure 5C*). However, *Mettl3null* heterozygosity has the inverse effect of suppressing each of these defects in *Nab2ex3* females: *Nab2ex3,Mettl3null/+* mutant females show approximately 3-fold higher viability (*Figure 5A*), 6-fold higher climbing rates (at the 30 s time point; *Figure 5B*), and 1.75-fold longer lifespan (*Figure 5C*) than *Nab2ex3* mutant females. As both Nab2 and Mettl3 act within the *Drosophila* nervous system (*Bienkowski et al., 2017*; *Lence et al., 2016*; *Kan et al., 2017*; *Kan et al., 2021*; *Corgiat et al., 2021*; *Rounds et al., 2021*), we sought to test whether this rescue of *Nab2ex3* by reduced *Mettl3* stems from cell autonomous roles for both factors within neurons. To address this hypothesis, we expressed a *UAS-Mettl3-RNAi* transgene in *Nab2ex3* neurons using the pan-neuronal

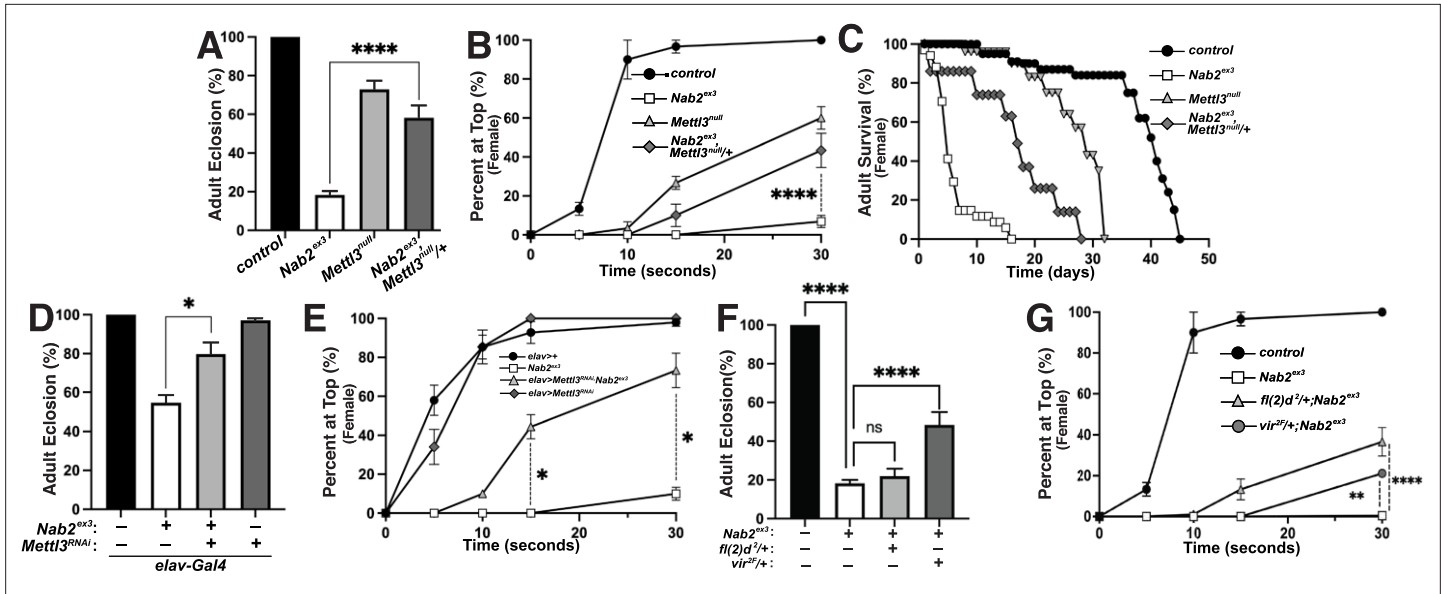

**Figure 5.** Reduction of the Mettl3 m⁶A transferase suppresses viability and behavioral defects in *Nab2* mutant females. (**A**) Percent of control, *Nab2^ex3^*, and *Nab2^ex3^,Mettl3^null/+^* flies that eclose as viable adults (calculated as #observed/#expected). (**B**) Negative geotaxis of age-matched adult females of the indicated genotypes over time in seconds. (**C**) Survival of age-matched adult female flies of the indicated genotypes over time in days. (**D**) Percent of *elav >Gal4* alone control, *elav-Gal4;;Nab2^ex3^*, *elav-Gal4;UAS-Mettl3-RNAi;Nab2^ex3^*, and *elav-Gal4;UAS-Mettl3-RNAi* flies that eclose as viable adults (calculated as #observed/#expected). Note that baseline *Nab2^ex3^* viability is elevated in the background of the *elav-Gal4* transgene, and significantly suppressed by inclusion of *UAS-Mettl3* RNAi. (**E**) Negative geotaxis assay for age-matched adult females of the indicated genotypes over time in seconds. (**F**) Percent of control, *Nab2^ex3^*, *Nab2^ex3^,fl(2)d^2/+^*, or *Nab2^ex3^,vir^2F/+^* flies that eclose as viable adults (calculated as #observed/#expected). (**G**) Negative geotaxis of age-matched adult females of the indicated genotypes over time in seconds. Significance values are indicated (*p<0.05, **p<0.01, ****p<0.0001).

The online version of this article includes the following figure supplement(s) for figure 5:

**Figure supplement 1.** Genomic PCR confirms the *Nab2^ex3^,Mettl3^null^* recombinant.

driver *elav-Gal4* (**Lin and Goodman, 1994**) (i.e., *elav-Gal4;UAS-Mettl3-RNAi;Nab2^ex3^*). Notably, this depletion of Mettl3 only in neurons was sufficient to suppress *Nab2^ex3^*-associated defects in both viability (**Figure 5D**) and locomotion (**Figure 5E**) in female flies, consistent with a functional interaction between Nab2 and Mettl3 in neurons. Similarly, we reasoned that reducing other components of the m6A 'writer' complex could rescue *Nab2^ex3^* defects. Indeed, we observed that heterozygous loss of two other components of the methyltransferase complex required for *Sxl* exon 3 skipping, *female-lethal(2)d* (*fl(2)d*) and *virilizer* (*vir*) (**Granadino et al., 1990**; **Hilfiker et al., 1995**; **Niessen et al., 2001**), also suppresses *Nab2^ex3^* mutant phenotypes (**Figure 5F and G**). Heterozygous loss of *fl(2)d* in a *Nab2^ex3^* mutant suppresses defects in female locomotion but not viability, while heterozygous loss of *virilizer* dominantly suppresses defects in both viability and locomotion.

## Nab2 binds *Sxl* pre-mRNA and modulates m⁶A methylation

The finding that reduced Mettl3 expression rescues viability, lifespan, and locomotor defects in *Nab2^ex3^* females indicates that the Mettl3 m⁶A 'writer' complex is required to promote developmental and behavioral defects in Nab2 mutants. However, loss of the same Mettl3 m⁶A 'writer' complex normally causes developmental and behavioral defects that resemble Nab2 mutant phenotypes documented here (**Figures 4 and 5**), and that are accompanied by *Sxl* exon 3 inclusion due to hypomethylation of *Sxl* mRNA (**Haussmann et al., 2016**; **Lence et al., 2016**; **Kan et al., 2017**; **Kan et al., 2021**). This paradox could be explained if *Sxl* exon 3 inclusion in *Nab2^ex3^* females accumulate excess m⁶A on the *Sxl* pre-mRNA. To test this hypothesis, a series of primer sets were designed to examine *Sxl* pre-mRNA and mRNA transcripts by RNA-immunoprecipitation (RIP) and anti-m⁶A-RIP (MeRIP) (**Figure 6A**). As illustrated in **Figure 6A**, the *Sxl* transcript contains candidate binding sites for both Sxl protein (polyuridine tracts = red ticks) and the polyadenosine RNA binding protein Nab2 protein (AC(A)₁₃ tract = green tick). Approximate mapped sites of m⁶A methylation (yellow ticks) in the



**Figure 6.** Nab2 associates with the *Sxl* mRNA and inhibits its m6A methylation. (**A**) Diagram of exons (E2, E3, E4) and introns (I2 and I3) of the *Sxl* pre-mRNA annotated to show coding sequence (CDS; black), the retained intronic region in *Nab2ex3* female (gray), and location of color-coded primer pairs (E2-F(orward) and E2-R(everse), E2-F and E4-R, I2-F and E3-R, I3-F, and E4-R), poly(U) sites (red lines), AC(A)13 site (green line), and mapped m6A positions in *Drosophila* heads (yellow lines) (**Kan et al., 2021**). (**B**) RT-qPCR analysis of *Act5c* and *Sxl* mRNA present in anti-m6A precipitates of control (*control*;

Figure 6 continued

black), *Nab2^ex3^* (white), or *Mettl3^null^* (gray) female heads. The position of *Sxl* primer pairs is indicated (E2-F+E2-R and E2-F+E4-R). (C) Similar analysis as in (B) using I2-F+E3-R and I3-F+E4-R primer pairs to detect unspliced *Sxl* transcripts in anti-m6A precipitates of control (black), *Nab2^ex3^* (white), or *Mettl3^null^* female heads. (D) Sanger sequencing traces showing C-to-U editing adjacent to m6A sites in control and *Nab2^ex3^* female head RNA samples subjected to DART-sanger sequencing (**Meyer, 2019**) within the retained intronic region of *Sxl* pre-mRNA. m6A sites are indicated by red asterisks. (E) Table of the m6A sites (red = hypermethylated in *Nab2^ex3^*, blue = no change in *Nab2^ex3^*) mapped by DART-sanger sequencing in (D) with the corresponding location (dm6), average C-to-U editing fraction (%), and ratio of C-to-U editing for *Nab2^ex3^* to control female head samples. Data are representative of three biological replicates. (F) Schematic showing the location of the m6A sites mapped by DART within exons (E3 and E4) and intron 3 of *Sxl* pre-mRNA. Site numbering corresponds to numbering in (E). (G) RT-qPCR analysis with the I3-F+E4-R primer pair in (A) from anti-Flag precipitates of control and *elav-Gal4,UAS-Nab2:Flag* female heads. (H) Top: schematic of the exon3-exon4 region of *Sxl* mRNA showing the intron region retained in *Nab2^ex3^* (gray fill) and the normal exon 3-exon 4 splicing product (green fill) and the aberrant exon 3-intron 3^503^-exon 4 (red-gray fill). Bottom: RT-PCR analysis of *Sxl* using the E3-E4 primer pair and RNAs harvested from female heads of the indicated genotypes: *elav-Gal4* alone, *elav-Gal4+Nab2^ex3/+^*, *Nab2^ex3^* mutant, *Nab2^ex3/+^*, *UAS-Mettl3* alone, or *elav>Mettl3+Nab2^ex3/+^*. Arrowheads denote exon 3-exon 4 and exon 3-intron 3^503^-exon 4 bands are indicated. Small gray arrow indicates Nab2-dependent splice variant. Asterisk marks a nonspecific band. (I) Percent of control, *Nab2^ex3^*, or *Nab2^ex3^;Ythdc1^ΔN^/+* flies that eclose as viable adults (calculated as # observed/# expected).

The online version of this article includes the following source data and figure supplement(s) for figure 6:

**Source data 1.** Raw agarose gel showing RT-PCR analysis of *Sxl* mRNA in the relevant genotypes shown in **Figure 6H**.

**Source data 2.** Raw agarose gel showing RT-PCR analysis of *RpL32* mRNA in the relevant genotypes shown in **Figure 6H**.

**Figure supplement 1.** Detailed schematic of the exon 2-3-4 *Sxl* locus with annotated locations of introns and exons annotated to show coding sequence (CDS; blue), the retained intronic region in *Nab2^ex3^* females (gray), and locations of color-coded primer pairs (E2-F and E2-R, E2-F and E4-R, I2-F, and E3-R, I3-F, and E4-R), poly(U) sites red lines, poly(A) sites green lines, and mapped m6A locations in *Drosophila* embryos yellow lines (**Kan et al., 2017**).

**Figure supplement 2.** Nab2 limits m6A methylation of additional RNAs.

**Figure supplement 3.** Heterozygosity for Mettl3 does not alter *Sxl* splicing in *control* or *Nab2^ex3^* mutant female heads.

**Figure supplement 3—source data 1.** Raw agarose gel showing RT-PCR analysis of *Sxl* mRNA in the relevant genotypes shown in **Figure 6—figure supplement 3**.

**Figure supplement 3—source data 2.** Raw agarose gel showing RT-PCR analysis of *RpL32* mRNA in the relevant genotypes shown in **Figure 6—figure supplement 3**.

**Figure supplement 4.** Neuronal overexpression of *Mettl3* RNA using the Gal4/UAS system.

*Drosophila* head transcriptome are also indicated (**Kan et al., 2021**; see **Figure 6—figure supplement 1** for a more detailed schematic).

To assess the m6A status of total *Sxl* RNA, MeRIP precipitates from female head lysates (control, *Nab2^ex3^*, and *Mettl3^null^*) were analyzed by reverse transcription-real time quantitative PCR (RT-qPCR) with the exon 2-exon 2 (E2-E2) primer pair, which amplifies both pre-mRNA and mature mRNA (*Sxl^E2-E2^* in **Figure 6B**). This approach detects reduced *Sxl* m6A in *Mettl3^null^* heads relative to controls, which is consistent with prior studies (**Haussmann et al., 2016**; **Lence et al., 2016**; **Kan et al., 2017**; **Kan et al., 2021**), and an increase in *Sxl* transcript recovered from MeRIP of *Nab2^ex3^* heads, consistent with increased *Sxl* m6A modification. As additional controls for m6A status, two m6A-methylated Mettl3-target RNAs, *Act5c* and *Usp16* (**Lence et al., 2016**; **Kan et al., 2017**; **Kan et al., 2021**) were analyzed. MeRIP-qPCR indicates that both mRNAs show decreased m6A in *Mettl3^null^* and show increased m6A in *Nab2^ex3^* flies (**Figure 6—figure supplement 2**). Shifting this analysis to qPCR with the *Sxl* E2-E4 primer set (*Sxl^E2-E4^* in **Figure 6B**), which detects spliced *Sxl* mRNA, reveals a similar pattern of elevated *Sxl* m6A in *Nab2^ex3^* heads. Together, these MeRIP-qPCR data argue that Nab2 either inhibits Mettl3-mediated m6A deposition or promotes m6A removal from *Sxl* mRNA. A prediction of this model is that Nab2 loss should result in increased levels of m6A on *Sxl* pre-mRNA. MeRIP analysis using the I2-E3 primer pair (*Sxl^I2-E3^* in **Figure 6C**) or the I3-E4 primer pair (*Sxl^I3-E4^* in **Figure 6C**) reveals moderate (1.5-fold) enrichment for intron 2-containing *Sxl* RNAs in *Nab2^ex3^* heads, and stronger (4.5-fold) enrichment for intron 3-containing RNAs, consistent with elevated m6A on *Sxl* pre-mRNAs that contain introns 2 and 3.

To more precisely define how loss of Nab2 alters the relative abundance and/or location of m6A deposition along the *Sxl* transcript in female heads, we utilized in vitro DART-Sanger sequencing (Deamination adjacent to RNA modification Targets followed by Sanger sequencing) (**Meyer, 2019**; **Tegowski et al., 2022**; **Figure 6D–F**). This method overcomes several limitations of traditional antibody-based methods including limited sensitivity and selectivity, and struggle to distinguish m6A from other RNA

modifications (i.e., m6Am) (**Meyer, 2019**). Briefly, in vitro DART-Sanger sequencing involves incubating RNA with a chimeric fusion protein consisting of the deaminating enzyme APOBEC1 fused to the m6A-binding YTH domain of m6A 'reader' proteins. As m6A-modified adenosine (A) residues are followed by a cytosine (C) residue in the most common consensus sequence (**Meyer, 2019**; **Linder et al., 2015**; **Meyer et al., 2012**), the APOBEC-YTH fusion recognizes m6A-modified A and deaminates the neighboring C, creating a uracil (U) base, which is read as a thymine (T) during Sanger sequencing. Therefore, C-to-U transitions and the frequency at which they occur permit mapping of m6A location and relative abundance. Thus, this method enables us to define the m6A modification status of the *Sxl* transcript in control and *Nab2ex3* heads. For this experiment, we treated RNA extracted from female control or *Nab2ex3* heads with APOBEC1-YTH, and subsequently performed RT-PCR with primers that amplify *Sxl* exon 3-intron 3503 (E3-I3 as illustrated in **Figure 6F**). Sanger sequencing and subsequent analysis of C-to-U transitions revealed the presence of seven m6A sites (sites #1–7) within this region (**Figure 6D–F**, denoted by asterisk in **Figure 6D**). These sites fall within or adjacent to sites mapped in a previous study of *Drosophila* head RNAs (**Kan et al., 2021**). Of the seven m6A modifications mapped within this region, four sites show a statistically significant increases C-to-U transition in *Nab2ex3* female heads compared to control (**Figure 6D**; four sites are denoted by red color in **Figure 6E–F**). Specifically, sites 1–3 and 7 show 2.00 ratio of m6A modification (calculated as %C-to-U *Nab2ex3*/%C-to-U *control*), providing evidence that these sites are methylated to a greater extent in *Nab2ex3* female heads compared to control heads. Notably, the m6A modification mapped to site 7 falls within the first adenosine residue of the proposed Nab2 $AC(A)_{13}$ binding site (see schematic in **Figure 6A and F**). These results are consistent with a role for Nab2 in inhibiting m6A levels on *Sxl* pre-mRNA and suggest that modulation of m6A levels may link Nab2 to other RNA targets within the *Drosophila* head transcriptome. To test whether Nab2 physically associates with *Sxl* pre-mRNA as a potential mechanism to limit m6A levels, an anti-Flag IP of FLAG-Nab2 was performed from head lysates of adult females expressing N-terminally tagged Nab2 specifically in neurons (*elav>Flag:Nab2*). RT-qPCR of precipitates analyzed with *Sxl* I3-E4 primers provides evidence that Nab2 associates with unspliced *Sxl* pre-mRNA (**Figure 6G**). In sum, these data provide a molecular framework to interpret *Nab2-Mettl3-Sxl* genetic interactions in which Nab2 associates with the *Sxl* pre-mRNA, perhaps via the $AC(A)_{13}$ site located in I3 (green tick; **Figure 6A**) and limits levels of m6A on this transcript.

In light of these m6A data, we revisited the effect of altered *Mettl3* gene dosage on *Sxl* RNA structure. Reducing *Mettl3* levels by half (*Mettl3null/+*) does not significantly alter *Sxl* splicing patterns in either control females or *Nab2ex3* females (**Figure 6—figure supplement 3**). Because complete removal of Mettl3 is lethal in animals that also lack Nab2, we considered whether overexpressing Mettl3 is sufficient to reproduce *Sxl* splicing defects we observe in *Nab2ex3* female heads. To test this possibility, we compared patterns of *Sxl* splicing between *Nab2ex3* female heads and heads from animals overexpressing Mettl3 in neurons using a *UAS-Mettl3* (**Lence et al., 2016**) transgene driven by *elav-Gal4* (**Lin and Goodman, 1994**; **Figure 6—figure supplement 4**). As shown in **Figure 6H**, RT-PCR using primers located in *Sxl* exon 3 and exon 4 detect mis-spliced exon 3-exon 4 RNA (green boxes) and the aberrant exon 3-intron 3503-exon 4 (red-gray boxes) product in *Nab2ex3* homozygous heads, with none or very low levels of these two RNA species in control (*elav-Gal4* alone) and *Nab2ex3* heterozygous heads. However, overexpression of Mettl3 in neurons is sufficient to produce the exon 3-exon 4 and exon 3-intron 3503-exon 4 RNAs in both control and *Nab2ex3* heterozygote heads, thus replicating the effect of Nab2 loss on *Sxl* splicing. This analysis also identified a *Sxl* exon 3-exon 4 splicing intermediate in female heads that is approximately 60 nt smaller than the expected exon 3-exon 4 product, which is lost in *Nab2ex3* female heads (**Figure 6H**, gray arrow). Sanger sequencing of this product revealed the presence of a Nab2-regulated cryptic 3′ splice site located within exon 3 that corresponds to the *Sxl-RZ*, *RK* and *RQ* RNAs (see FlyBase).

The increase in m6A levels detected on *Sxl* pre-mRNA upon loss of Nab2 provides evidence that Nab2 normally limits methylation on some RNAs. Excess m6A on transcripts in *Nab2ex3* heads could lead to over-recruitment of the nuclear m6A YTH-domain containing 'reader' protein, Ythdc1 (or YT521-B), which regulates nuclear processing of many pre-mRNA targets including the removal of *Sxl* exon 3 in females (**Haussmann et al., 2016**). Thus, we tested whether reducing levels of Ythdc1 with the *Ythdc1ΔN* null allele (**Lence et al., 2016**) could rescue the lethality of *Nab2ex3* mutants. Indeed, heterozygous loss of Ythdc1 increases viability of *Nab2ex3* females approximately fivefold (**Figure 6I**). This finding is consistent with biochemical evidence that Nab2 represses m6A levels on the *Sxl* RNA

and provides additional evidence that Nab2 interacts genetically with multiple elements of the m⁶A machinery.

## Discussion

Through an unbiased high-throughput RNA sequencing approach, we identify a set of head-enriched RNAs in *Drosophila* whose levels or structure are significantly affected by loss of the Nab2 RBP, with the latter effect on RNA structure traced to splicing defects (including intron retention, alternative 5′ and 3′ splice site usage, and exon skipping) in a small group of approximately 150 transcripts. The top-ranked Nab2-regulated splicing event is skipping of *Sxl* exon 3 in females, which prior studies *Haussmann et al., 2016*; *Lence et al., 2016*; *Kan et al., 2017* have shown to be guided by m⁶A methylation of specific sites in the *Sxl* pre-mRNA. Our biochemical studies reveal that Nab2 inhibits hypermethylation of sites in and around *Sxl* exon3, and genetic data show that developmental and behavioral phenotypes resulting from Nab2 loss are rescued by decreasing levels of the Mettl3 methyltransferase, other components of the Mettl3 complex, or the nuclear m⁶A reader protein Ythdc1. Data suggest that Nab2-Mettl3 coregulation of *Sxl* splicing is most significant in neurons – the effect of Nab2 on *Sxl* splicing is strongest in tissues that contain CNS components (e.g., brain and ventral nerve cord), while Mettl3 overexpression only in neurons is sufficient to replicate *Sxl* splicing defects seen in *Nab2* mutant heads. This apparent tissue specificity of the link between Nab2 and Mettl3 may help explain neurological defects in mice and humans lacking the Nab2 ortholog ZC13H14, although lethality of animals lacking both Nab2 and Mettl3 is consistent with only partial overlap between RNA targets of these two RBPs.

Because *Sxl* exon 3 contains a translational termination (stop) codon, inclusion of this exon disrupts female-specific expression of Sxl protein, a U-rich RNA binding protein that controls somatic and germline sexual identity via effects on splicing and translation of target mRNAs (rev. in *Penalva and Sánchez, 2003*; *Moschall et al., 2017*). Multiple lines of evidence suggest that *Sxl* mRNA may be a particularly significant target of Nab2 in neurons: mis-spliced RNAs in *Nab2* mutant female heads are enriched for bioinformatically predicted Sxl binding motifs, and the *Sxl^{M8}* allele that constitutively skips exon 3 (*Barbash and Cline, 1995*) substantially reverses developmental and behavioral defects in *Nab2* null females (*Figure 4*). Moving downstream of Sxl, alleles of male-specific DCC components, including the direct Sxl target *msl-2* (*Bashaw and Baker, 1995*; *Bashaw and Baker, 1997*), also rescue phenotypic defects in *Nab2* mutant females (*Figure 4* and *Figure 4—figure supplement 1*). Given that these DCC components are not normally expressed or active in females, these data provide evidence that masculinized *Sxl* splicing and DCC activity contribute to developmental and behavioral defects in *Nab2* mutant female flies. Elevated DCC activity could contribute to axon projection defects in female MBs, but this seems unlikely given that *Nab2^{ex3}* males develop similar MB axonal defects (*Kelly et al., 2016*). Overall, these data imply a specific link between Nab2 and the *Sxl* exon 3 splicing machinery, which is confirmed by strong genetic interactions between *Nab2* and the *Mettl3* methyltransferase that promotes exon 3 skipping by depositing m⁶A on *Sxl* pre-mRNA (*Haussmann et al., 2016*; *Lence et al., 2016*; *Kan et al., 2017*).

Molecular assays provide key insight into the Nab2-Sxl interaction. A tagged form of Nab2 protein associates with unspliced *Sxl* pre-mRNA when expressed in brain neurons, and Nab2 loss results in excess m⁶A on *Sxl* mRNA as detected by two independent assays used to map m⁶A sites, meRIP-qPCR and DART. The high resolution of the DART technique allowed us to map m⁶A sites in the *Sxl* exon 3-intron 3-exon 4 region that are more highly methylated in *Nab2* mutants than in controls, consistent with Nab2 inhibiting m⁶A accumulation at sites normally modified by the Mettl3 complex. Significantly, these Nab2-regulated methylation sites lie under or adjacent to anti-m⁶A-CLIP peaks mapped in the *Sxl* RNA from adult female heads (*Kan et al., 2021*) and thus complement and extend our knowledge of m⁶A patterns on *Sxl* mRNAs expressed in the adult head. Given the known role of m⁶A in promoting *Sxl* exon 3 excision (*Haussmann et al., 2016*; *Lence et al., 2016*; *Kan et al., 2017*), these data collectively support a model in which Nab2 interacts with the *Sxl* pre-mRNA in the nucleus and opposes m⁶A methylation by the Mettl3 complex, thus ensuring a level of m⁶A necessary to guide *Sxl* exon 3 skipping in the female nervous system. We term this a 'goldilocks' model in which either too little or too much m⁶A methylation of the region surrounding *Sxl* exon 3 can result in its retention in the developing female brain. These data provide the first evidence that the highly conserved Nab2

RBP is a key regulator of splicing in the adult brain, and that Nab2 is required to limit m⁶A modification of an RNA.

Studies employing the *Sxl^M8* allele indicate that altered *Sxl* splicing and decreased Sxl protein contribute to *Nab2* mutant phenotypes in females. As Sxl is itself an RBP that can control splicing, some fraction of the mis-sliced mRNAs detected by *Nab2^ex3* high-throughput sequencing may thus be Sxl targets. This hypothesis is supported by the substantial rescue conferred by the *Sxl^M8* allele and the enrichment for candidate Sxl-binding sites among mis-spliced mRNAs in *Nab2* mutant female heads. However, splicing of the Sxl target RNA *tra* is unaffected in the *Nab2* mutant RNA-Seq data-sets. The lack of effect on *tra* could be due to lack of read depth in the RNA-seq data, although this does not seem to be the case (see *Figure 4—figure supplement 1*), or to alternative Sxl target RNAs in adult heads. Unbiased screens for Sxl target RNAs have carried out in ovaries (*Primus et al., 2019*) and primordial germ cells (*Ota et al., 2017*), but a similar approach has not been taken in the adult nervous system, where Sxl targets may differ from other tissue types. In this regard, the group of Nab2-regulated RNAs identified here may be enriched for neuronal RNAs that are directly regulated by Sxl.

Although this study focuses on *Sxl* as a female-specific Nab2 regulated RNA, a large majority of other mis-splicing events in *Nab2* mutant head RNAs occur in both males and females. This evidence of a Nab2 role in non-sex-specific splicing events parallels evidence of accumulation of ~100 intron-containing pre-mRNAs in *nab2* mutant *S. cerevisiae* cells (*Soucek et al., 2016*). Rescue of *Nab2* mutant males and females by neuron-restricted expression of human ZC3H14 (*Kelly et al., 2014*) implies that this specificity may be a conserved in ZC3H14 proteins in higher eukaryotes. Indeed, knockdown of ZC3H14 in cultured vertebrate cells results in pre-mRNA processing defects and intron-specific splicing defects in the few RNAs that have been examined (*Morris and Corbett, 2018*; *Wigington et al., 2016*). The basis for Nab2 target specificity in *Drosophila* heads is not clear but could be due selectivity in binding to nuclear pre-mRNAs (e.g., *Sxl*) or interactions between Nab2 and partner proteins that define splicing targets.

Site-specific hypermethylation of *Sxl* resulting from Nab2 loss could arise by several mechanisms, including Nab2 modulating m⁶A deposition by blocking access of the Mettl3 complex to its target sites, or to Nab2 recruitment of an m⁶A 'eraser'. However, recent studies demonstrating that Nab2 and ZC3H14 each co-purify at nearly stoichiometric levels with the exon junction complex (EJC) (*Obrdlik et al., 2019*; *Singh et al., 2012*) and that the EJC binding locally excludes Mettl3-mediated m⁶A deposition on pre-mRNAs (*He et al., 2023*; *Uzonyi et al., 2023*; *Yang et al., 2022*) suggest an alternate model in which Nab2 inhibits m⁶A deposition in cooperation with the EJC. Notably, the human homolog of the *Drosophila* protein Virilizer, which is an m⁶A methyltransferase subunit and splicing factor (*Hilfiker et al., 1995*; *Niessen et al., 2001*), was recovered in an IP/mass-spectrometry screen for ZC3H14 nuclear interactors (*Morris and Corbett, 2018*). This finding raises an additional possibility that ZC3H14/Nab2 modulates m⁶A methylation via interactions with both the Mettl3 complex and the EJC. Moreover, evidence that m⁶A modulatory role of Nab2 is not restricted to the *Sxl* mRNA (see *Figure 6B* and *Figure 6—figure supplement 2*) raises the additional hypothesis that changes in abundance or structure of the group of Nab2-regulated RNAs defined in this study (see *Figures 1 and 2*) are due in part to changes in m⁶A status.

Prior work has shown that almost all developmental and behavioral defects caused by Nab2 loss can be traced to a Nab2 role within central nervous system neurons (*Pak et al., 2011*; *Kelly et al., 2014*; *Kelly et al., 2016*; *Corgiat et al., 2022*; *Rounds et al., 2021*). Suppression of these phenotypes by heterozygosity for *Sxl^M8* or *Mettl3^null* alleles or by neuron-specific *Mettl3* RNAi is thus consistent with a mechanism in which Nab2 inhibits steady-state m⁶A levels on a group of neuronal RNAs, and that *Sxl* is one of these RNAs in the female brain. However, the lack of statistically significant rescue of *Sxl* splicing defects by *Mettl3* heterozygosity (*Figure 6—figure supplement 3*) implies that *Sxl* splicing is only rescued in a small subset of cells or that *Sxl^M8* and *Mettl3^null* heterozygosity rescue *Nab2* mutant phenotypes through different mechanisms. While *Sxl^M8* specifically restores a single splicing event in a single mRNA, the *Mettl3^null* allele has the potential to broadly affect m⁶A levels on multiple RNAs with subsequent effects on multiple m⁶A-dependent processes in the cytoplasm, including mRNA export to the cytoplasm and translation. One potential candidate mRNA of this type is *Wwox*, which encodes a conserved WW-domain oxidoreductase that accumulates in brains of *Nab2* mutant flies (*Corgiat et al., 2021*) and is mutated in human spinocerebellar ataxia type 12 (*Serin et al., 2018*; *Mallaret*

*et al., 2014*). Significantly, the *Wwox* RNA has a 3′UTR intron that is retained in *Nab2* mutant heads (this study) and contains a candidate m⁶A site (*Kan et al., 2021*), suggesting that *Wwox* RNA may be a target of both Nab2 and Mettl3. Elevated Wwox protein is also detected in the hippocampus of *ZC3H14* knockout mice, raising the possibility that Nab2 and ZC3H14 share some common RNA targets across species (*Roxas et al., 2017*). ZC3H14 has to date not been linked to the m⁶A mark in mouse or human cells. However, the enrichment for *Sxl* mis-splicing in neuronal tissue (see *Figure 3F*) and rescue by *ZC3H14* when expressed in neurons of otherwise *Nab2*-deficient animals (*Kelly et al., 2016*) supports the hypothesis that the Nab2/ZC3H14 family of RBPs may share an m⁶A inhibitory role that is specific to neurons, and that excessive m⁶A methylation of RNAs also contributes to neurological deficits in mice and humans lacking ZC3H14.

## Materials and methods

### *Drosophila* stocks and genetics

*Drosophila melanogaster* stocks and crosses were maintained in humidified incubators at 25°C with 12 hr light-dark cycles. The alleles *Nab2^ex3^* (null), *Nab2^pex41^* (*precise excision 41*; control), and *UAS-Flag-Nab2* have been described previously (*Pak et al., 2011*; *Kelly et al., 2014*). Lines from Bloomington Drosophila Stock Center (BDSC): *GMR-Gal4* (#1350), *elav^C155^-Gal4* (#458), *msl-2^227^* (#5871), *msl-2^kmA^* (#25158), *mle^9^* (#5873), *roX1^ex6^* (#43647), *UAS-Mettl3, UAS-Mettl3-RNAi* (#80450), *fl(2)d^2^* (#36302), *vir^2F^* (#77886). The *Mettl3^null^*, *UAS-Mettl3*, and *Ythdc^ΔN^* alleles were all kind gifts of J-Y. Roignant. The *Nab2^ex3^,Mettl3^null^* and *Nab2^ex3^, Ythdc^ΔN^* chromosomes were generated by meiotic recombination and confirmed by genomic PCR. A total of 200 recombinant lines were screened to identify *Nab2,Mettl3* double mutants.

### RNA sequencing (RNA-Seq) on *Drosophila* heads

RNA-Seq was performed on three biological replicates of 60 newly eclosed adult female and male *Drosophila* heads genotype (control and *Nab2^ex3^* mutants). Heads were collected on dry ice, lysed in TRIzol (Thermo Fisher), phase-separated with chloroform, and ran through a RNeasy Mini Kit purification column (QIAGEN). Samples were treated with DNase I (QIAGEN) to remove DNA contamination and transported to the University of Georgia's Genomics and Bioinformatics Core for sequencing. rRNA was depleted using a Ribo-Zero Gold Kit (Illumina) and cDNA libraries were prepared using a KAPA Stranded RNA-Seq Kit (Roche). Quality control steps included initial Qubit quantification along with RNA fragment size assessment on an Agilent 2100 Bioanalzyer before and after rRNA depletion. The cDNA libraries were then sequenced for 150 cycles on a NextSeq 500 High Output Flow Cell (Illumina) set to generate paired-end, 75 base-pair (bp) reads. Total sequencing yield across all samples was 81.48 Gbp, equivalent to about 1.1 billion reads in total and 91 million reads per sample. Sequencing accuracy was high; 93.52% of reported bases have a sequencing quality (Q) score ≥ 30.

### Read mapping, differential expression, and visualization

Raw read FASTA files were analyzed on the Galaxy web platform (usegalaxy.org; *Afgan et al., 2018*). The BDGP6 release *Drosophila melanogaster* genome (*dos Santos et al., 2015*) from release 92 of the Ensembl database (*Yates et al., 2020*) was used as input for subsequent read mapping, annotation, and visualization. Briefly, reads from all four NextSeq500 flow cell lanes were concatenated using the Galaxy *Concatenate datasets tail-to-head (cat)* tool and mapped using RNA STAR (*Dobin et al., 2013*) with default parameters with some modifications. For each Galaxy tool, version numbers and exact parameters used are detailed in the following table:

| Galaxy software and parameters | | |
|---|---|---|
| Tool | Concatenate datasets tail-to-head (cat) | Default parameters |
| | Galaxy version 0.1.0 | |
| Tool | RNA STAR | Default parameters with the following exceptions: |

*Continued on next page*

*Continued*

| | Galaxy software and parameters | |
|---|---|---|
| | Galaxy version 2.5.2b-0 | Read type: paired |
| | | Reference genome: *from history (using Ensembl FASTA and GTF referenced in text)* |
| Tool | featureCounts | Default parameters with the following exceptions: |
| | Galaxy version 1.6.0.3 | Gene annotation file: *history (Ensembl GTF referenced in text)* |
| | | Count fragments instead of reads: *enabled* |
| | | GFF gene identifier: *gene_name* |
| | | Strand specificity: *stranded-reverse* |
| Tool | DESeq2 | Default parameters with the following exceptions: |
| | Galaxy version 2.11.40.1 | *Factors: four levels, each a group of three biological replicates* |
| | | Output normalized counts table – *true* |
| | | Output all levels vs. all levels – *true* |
| Tool | DEXSeq-Count | Default parameters with the following exceptions: |
| | Galaxy version 1.20.1 | In 'read count' mode: strand-specific library – yes, reverse |
| Tool | DEXSeq | Default parameters with the following exception: |
| | Galaxy version 1.20.1 | Visualize results? – *no* |
| | Gene Ontology (GO) software and parameters | |
| Tool | GO2MSIG | Parameters: |
| | web interface | Data source: *NCBI gene2go* |
| | | Taxon ID – 7227 |
| | | Evidence codes: *include EXP, IDA, IEP, IGI, IMP, IPI, ISS, TAS* |
| | | Propagate associations – *true* |
| | | Use gene – *symbol* |
| | | Repress IDs – *no* |
| | | Create genesets for – *[1 top-level domain only]* |
| | | Max. geneset size – *700* |
| | | Min. geneset size – *15* |
| | | Output format – *gmt* |
| | | Database release – *April 2015* |
| Tool | GSEA Desktop for Windows | Default parameters with the following exceptions: |
| | v4.0.3 | *For up- and downregulated transcripts in Nab2^{ex3} vs. control:* |
| | GSEA-Preranked | zip-report – *true* |
| | | plot_top_x – *100* |

*Continued on next page*

| Continued | | | |
|---|---|---|---|
| Galaxy software and parameters | | | |
| | | create_svgs – *true* | |
| | | Collapse – *No collapse* | |
| | | *For alternatively spliced transcripts in Nab2^{ex3} vs. control:* | |
| | | zip-report – *true* | |
| | | Minimum gene set size – 5 | |
| | | create_svgs – *true* | |
| | | Collapse – *No collapse* | |
| Tool | AmiGO 2 | Default parameters | |
| | web interface | | |

Mapped reads were assigned to exons and tallied using featureCounts (**Liao et al., 2014**) default parameters with some modifications noted above. Differential expression analysis was conducted for all 12 samples using DESeq2 (**Love et al., 2014**; Galaxy version 2.11.40.1) and default parameters with some modifications noted above. Differential exon usage was analyzed using Galaxy version 1.20.1 of DEXSeq (**Anders et al., 2012**) and the associated Galaxy tool DEXSeq-Count in both '*prepare annotation*' and '*count reads*' modes. Both tools were run with the Ensembl GTF with default parameters with some modifications noted above. Unlike with DESeq2, female samples and male samples were compared in independent DEX-Seq analyses. Outputs from all tools were downloaded from Galaxy for local analysis, computation, and visualization.

Custom R scripts were written to generate volcano plots and heatmaps. Additional R packages used include ggplot2 (**Wickham, 2016**) and ggrepel (**Slowikowski, 2019**). R scripts were written and compiled in RStudio (**RStudio Team, 2018**). Principal component analysis was conducted on Galaxy. Mapped reads were visualized in the IGV (**Robinson et al., 2017**) and annotated based on data available on FlyBase (**Thurmond et al., 2019**). Significant fold change values in either male or female from DESeq2 (adj. p-val<0.05 and |log$_2$FC| > 1) were plotted, with the color indicating the fold change threshold reached in either males or females. Significantly DE genes (adj. p-val<0.05 and |log$_2$FC| > 1) were classified by type, as indicated by their gene ID.

## MISO analysis

MISO (**Katz et al., 2010**) version 0.5.4 was used to determine percent spliced in (PSI) values for annotated alternative 3′ splice sites, alternative 5′ splice sites, and retained introns for each sample separately as follows. Alternative splicing annotations were generated using the rnaseqlib (a direct link to script is provided at https://rnaseqlib.readthedocs.io/en/clip/) script, gff_make_annotation.py, with `flags--flanking-rule` commonshortest `--genome-label` dm6. Replicates for each sample were pooled, and only full-length, mapped reads (76 bp) were used for the MISO analysis as MISO requires all reads input to be of the same length. MISO was run with the flag-prefilter, and the output was then input into the script, summarize_miso, with the flag `--summarize-samples`. Next, differential, alternative 5′ and 3′ splice sites, and differential retained introns, were determined comparing *Nab2^{ex3}* and control for males and females, separately, using the script, compare_miso, with flag `--compare-samples`. The output of compare miso was then input into the script, filter_events, with the flags `--filter --num-inc` 10 `--num-exc` 10 `--num-sum-inc-exc` 50 `--delta-psi` 0.3 `--bayes-factor` 10, to obtain the final differential PSI values.

## GO analysis

GSEA software (**Subramanian et al., 2005**) was employed for GO analysis (**The Gene Ontology Consortium, 2019**). For clarity, analyses were conducted separately for each of the three top-level GO domains: *molecular function*, *biological process*, and *cellular component*. GSEA-compatible GO term gene sets for *D. melanogaster* were acquired using the GO2MSIG web interface (**Powell, 2014**). GSEA Desktop for Windows, v4.0.3 (Broad Institute) was then used to identify two distinct classes of GO terms, independently for females and males: (1) terms enriched among up- and downregulated

transcripts in *Nab2ex3* compared to controls, and (2) terms enriched among transcripts alternatively spliced in *Nab2ex3* compared to controls. For the first class, inputs consisted of all genes whose expression could be compared by DESeq2 (i.e., adjusted p-value ≠ NA). For the second class, inputs consisted of all genes with previously annotated alternative splicing events according to MISO. To identify the first class of GO terms, genes were ranked by $\log_2$ (fold change) calculated by DESeq2 and analyzed by the GSEA-Pre-ranked tool. To identify the second class of GO terms, genes with were ranked by the absolute value of the difference in PSI comparing *Nab2ex3* and control calculated by MISO. This second ranking was analyzed by the GSEA-Preranked tool. Enriched GO terms (nominal p-value<0.05) identified for the first class were evaluated manually, surfacing multiple terms directly related to splicing. Enriched GO terms (nominal p-value<0.05) for the second class were ordered by normalized enrichment score (NES) and evaluated to identify the top 'independent' GO terms. Terms were defined as 'independent' by reference to their position in the GO hierarchy as reported on each term's 'Inferred Tree View' page of the AmiGO2 GO database web tool (*Carbon et al., 2009*). 'Independent' terms had no parent, child, or sibling terms in the GO hierarchy associated with a higher NES than their own.

## RBPs motif enrichment analysis using MISO analysis

RNA sequences were analyzed at differentially retained introns and alternative 3′ and 5′ splice sites obtained from the MISO analysis on males and females separately (*Nab2ex3* mutants vs. control). The sequence for each of these went 25 bp into the exon(s) of interest and 1 kb into the intron of interest. In the case of alternative 3′ and 5′ splice sites, the sequences went 25 bp into the exon starting from the alternative spice site that is closest to the center of the exon (i.e., the inner-most splice site), and 1 kb into the intron starting from that inner-most spice site. To convert these to RNA sequences, DNA sequences were first obtained using fastaFromBed (*Quinlan and Hall, 2010*), and then all T's were converted to U's with a custom script. To obtain putative binding sites for RBPs at these sequences, the sequences were then input into fimo using the flags `--text --max-strand` and the 'Ray2013_rbp_*Drosophila_melanogaster*.meme' file (*Grant et al., 2011*).

## RNA isolation for RT-PCR and real-time qPCR

Total RNA was isolated from adult tissues with TRIzol (Invitrogen) and treated with DNase I (QIAGEN). For RT-PCR, cDNA was generated using SuperScript III First Strand cDNA Synthesis (Invitrogen) from 2 µg of total RNA, and PCR products were resolved and imaged on 2% agarose gels (Bio-Rad image). Quantitative real-time PCR (qPCR) reactions were carried out in biological triplicate with QuantiTect SYBR Green Master Mix using an Applied Biosystems StepOne Plus real-time machine (ABI). Results were analyzed using the ΔΔCT method, normalized as indicated (e.g., to *Act5C*), and plotted as fold-change relative to control.

## Primers used for RT and qPCR analysis

| Name | Sequence | Detects |
|---|---|---|
| *Sxl* pre-mRNA | Fwd: AGAACCAAAACTCCCTTACAGC<br>Rev: GTGAGTGTCTTTCGCTTTTCG | Intron 2-exon 3 |
| *Sxl* pre-mRNA | Fwd: ACCAATAACCGACAACACAATC<br>Rev: ACATCCCAAATCCACGCCCACC | Intron 3-exon 4 |
| *Sxl* mRNA | Fwd: GCTGAGCGCCAAAACAATTG<br>Rev: AGGTGAGTTTCGGTTTTACAGG | Exon 2-exon 2 |
| *Sxl* RT-PCR | Fwd: ACACAAGAAAGTTGAACAGAGG<br>Rev: CATTCCGGATGGCAGAGAATGG | Exon 2-3-4 |
| *Sxl* RT-PCR | Fwd: CTCTCAGGATATGTACGGCAAC<br>Rev: CATTCCGGATGGCAGAGAATGG | Exon 2-3-4 |

*Continued on next page*

*Continued*

| Name | Sequence | Detects |
|---|---|---|
| *Sxl* RT-PCR | Fwd: AGTATGTAGTTTTTATTTGCACGGG<br>Rev: CATTCCGGATGGCAGAGAATGG | Exon 3-4 |
| *Sxl* mRNA exon 2-4 transcript | Fwd: GATTGAATCTCGATCATCGTTC<br>Rev: CATTCCGGATGGCAGAGAATGG | Exon 2-exon 4 |
| *Sxl* mRNA exon 3-4 transcript | Fwd: CGAAAAGCGAAAGACACTCACTG<br>Rev: CATTCCGGATGGCAGAGAATGG | Exon 3-exon 4 |
| *Act5C* | Fwd: GAGCGCGGTTACTCTTTCAC<br>Rev: ACTTCTCCAACGAGGAGCTG | *Actin5C* |
| *USP-16-45-RF* | Fwd: ACACTTGGTCACGTCGTTCA<br>Rev: GGGCGCGCTCTTGAATTTAC | *USP-16* |

## Immunoblotting

For analysis of Sxl protein levels, *Drosophila* were reared at 25°C. Heads of newly eclosed flies were collected on dry ice. Protein lysates were prepared by homogenizing heads in 0.5 mL of RIPA-2 Buffer (50 mM Tris-HCl, pH 8; 150 mM NaCl; 0.5% sodium deoxtcholate; 1% NP40; 0.1% SDS) supplemented with protease inhibitors (1 mM PMSF; Pierce Protease Inhibitors; Thermo Fisher Scientific). Samples were sonicated 3 × 10 s with 1 min on ice between repetitions, and then centrifuged at 13,000 × *g* for 15 min at 4°C. Protein lysate concentration was determined by Pierce BCA Protein Assay Kit (Life Technologies). Head lysate protein samples (40–60 µg) in reducing sample buffer (50 mM Tris HCl, pH 6.8; 100 mM DTT; 2% SDS; 0.1% Bromophenol Blue; 10% glycerol) were resolved on 4–20% Criterion TGX Stain-Free Precast Polyacrylamide Gels (Bio-Rad), transferred to nitrocellulose membranes (Bio-Rad), and incubated for 1 hr in blocking buffer (5% non-fat dry milk in 0.1% TBS-Tween) followed by overnight incubation with anti-Sxl monoclonal antibody (1:1000; DHSB #M18) diluted in blocking buffer. Primary antibody was detected using species-specific horse-radish peroxidase (HRP) conjugated secondary antibody (Jackson ImmunoResearch) with enhanced chemiluminescence (ECL, Sigma).

## Viability and lifespan analysis

Viability at 25°C was measured by assessing eclosion rates of among 100 wandering L3 larvae collected for each genotype, and then reared in a single vial. Hatching was recorded for 5–6 d. At least three independent biological replicates per sex/genotype were tested and significance was calculated using grouped analysis on GraphPad (Prism). Lifespan was assessed at 25°C as described previously (*Morton et al., 2020*). In brief, newly eclosed animals were collected, separated by sex, placed in vials (10 per vial), and transferred to fresh vials weekly. Survivorship was scored daily. At least three independent biological replicates per vial of each genotype were tested, and significance was calculated using grouped analysis on GraphPad (Prism).

## Locomotion assays

Negative geotaxis was tested as previously described (*Morton et al., 2020*). Briefly, newly eclosed flies (day 0) were collected, divided into groups of 10 male or females, and kept in separate vials for 2–5 d. Cohorts of age-matched flies were then transferred to a 25 mL graduated cylinder for analysis. At least three biological replicates per sex were analyzed per genotype using GraphPad (Prism).

## Flag and m⁶A RNA immunoprecipitation (Flag-RIP and MeRIP)

The FLAG-RIP and MeRIP protocols were performed using previously described protocols (*Bienkowski et al., 2017*) and (*Lence et al., 2016*) with some modification. Briefly, three replicates of 30 newly eclosed female flies were collected in 1.5 mL Eppendorf tubes and frozen in dry ice. Heads were removed with a 5.5 Dumont tweezer and homogenized with a mortar/pestle in Isolation buffer (50 mM Tris-HCl pH 8.1, 10 mM EDTA, 150 mM NaCl, and 1% SDS, 50 mM NaCl). This preparation was diluted into IP buffer (50 mM HEPES, 150 mM NaCl, 5 mM EDTA, 0.5 mM DTT, 0.1% NP-40)

supplemented with protease inhibitors (Roche) and RNasin Plus Inhibitor (Promega). Lysates were incubated with anti-Flag (M2 clone; Sigma) or anti-m$^6$A (Synaptic Systems) antibody and recovered on magnetic Protein G Dynabeads (Invitrogen). After overnight incubation at 4°C with rocking, beads were washed 5× in IP buffer and RNA was isolated from antibody-bead precipitates, or controls (input samples) using TRIzol (Thermo Fisher). Samples were treated with DNase I and RNA was purified using RNeasy Kit (QIAGEN).

## Deamination adjacent to RNA modification targets (DART)

APOBEC-YTH and APOBEC-YTH$^{mut}$ were purified and in vitro DART-Sanger sequencing assays were performed as previously described (*Tegowski et al., 2022*) with minor modifications. Briefly, total RNA was isolated from adult heads with TRIzol (Invitrogen) and treated with DNase I (NEB). RNA was isolated once more with TRIzol (Invitrogen) to remove DNase I and DNase I Buffer (NEB). Next, 200 ng of purified RNA from *Drosophila* heads was incubated with 1000 ng of purified DART protein in DART buffer (10 mM Tris-HCl, pH 7.4, 50 mM KCl, 0.1 M ZnCl$_2$) and 1 μL of RNaseOUT (Invitrogen) in a total volume of 200 μL for 4 hr at 37°C. RNA was isolated with the QIAGEN Plus Micro Kit (QIAGEN) and stored at –80°C before being thawed for downstream Sanger sequencing analysis. cDNA was made using iScript Reverse Transcription Supermix (Bio-Rad). PCR amplification of *Sxl* pre-mRNA was carried out with Phusion High Fidelity PCR Kit (NEB). The resulting PCR product was PCR-purified using the QIAGEN PCR Purification Kit (QIAGEN). Samples were submitted for Sanger sequencing (McLabs) and %C-to-U editing was quantified using EditR software (*Kluesner et al., 2018*).

## Primers used for DART PCR and Sanger sequencing

| Name | Sequence | Detects |
|------|----------|---------|
| *Sxl* DART-PCR | Fwd: ACATATTTTTTTTCACAGCCCAG<br>Rev: TCAAAACGATCCCCCAGTTAT | Exon 3-intron 3 |
| *Sxl* DART Sanger Seq | Fwd: TTTTCACAGCCCAGAAAGAAGC | Exon 3-intron 3 |

## Statistical analysis

Group analysis on biological triplicate experiments was performed using two-way ANOVA (Turkey's multiple-comparison test) on GraphPad (Prism) version 8.4.2 (464). Sample sizes (n) and p-values are denoted in the text, figures, and/or figure legends and indicated by asterisks (e.g., *p<0.05).

## Acknowledgements

Stocks obtained from the Bloomington Drosophila Stock Center/BDSC (NIH P40OD018537) were used in this study. We thank members of the Moberg and Corbett Labs for helpful discussions and advice, and B Bixler, J Tanquary, and C Bowen for their contributions. We also thank M Alabady (Georgia Genomics and Bioinfomatics Core) for technical support, and T Lence and J-Y Roignant (Lausanne) for the gift of the *Mettl3* allele, and T Cline (Berkeley) for discussions and insights on the *Sxl[M8]* allele. This work was funded by grants from the National Institute of Health to KHM and AHC (R01 MH10730501), KDM (NIGMS RM1 HG011563; R01 MH118366), JCR (F31 HD088043), BJ (F31 NS103595), and CLL (F31 NS127545). BJ and BEB were also supported during portions of the study by the Emory Initiative to Maximize Student Development (NIH R25 GM125598).

# Additional information

### Funding

| Funder | Grant reference number | Author |
|--------|------------------------|--------|
| National Institutes of Health | R01 MH107305 | Ken Moberg<br>Anita Corbett |

| Funder | Grant reference number | Author |
|---|---|---|
| National Institutes of Health | F31 NS103595 | Binta Jalloh |
| National Institutes of Health | F31 HD088043 | J Christopher Rounds |
| National Institutes of Health | R25 GM125598 | Anita Corbett |
| National Institutes of Health | F31 HD079226 | Rick S Bienkowski |
| National Institutes of Health | F32 GM125350 | Derrick J Morton |
| National Institutes of Health | RM1 HG011563 | Kate Meyer |
| National Institutes of Health | R01 MH118366 | Kate Meyer |

The funders had no role in study design, data collection and interpretation, or the decision to submit the work for publication.

## Author contributions

Binta Jalloh, Conceptualization, Formal analysis, Supervision, Funding acquisition, Validation, Investigation, Visualization, Methodology, Writing - original draft, Writing – review and editing; Carly L Lancaster, Conceptualization, Data curation, Formal analysis, Supervision, Funding acquisition, Validation, Investigation, Methodology, Writing - original draft, Project administration; J Christopher Rounds, Conceptualization, Data curation, Software, Formal analysis, Supervision, Funding acquisition, Investigation, Visualization, Methodology, Writing – review and editing; Brianna E Brown, Formal analysis, Validation, Investigation; Sara W Leung, Resources, Formal analysis, Validation, Investigation, Visualization; Ayan Banerjee, Conceptualization, Investigation; Derrick J Morton, Conceptualization; Rick S Bienkowski, Conceptualization, Supervision, Investigation; Milo B Fasken, Conceptualization, Formal analysis, Supervision, Investigation, Visualization, Methodology; Isaac J Kremsky, Data curation, Software, Formal analysis, Visualization; Matthew Tegowski, Resources, Methodology; Kate Meyer, Resources, Funding acquisition, Methodology; Anita Corbett, Conceptualization, Supervision, Funding acquisition, Writing - original draft, Project administration, Writing – review and editing; Ken Moberg, Conceptualization, Supervision, Funding acquisition, Investigation, Writing - original draft, Project administration, Writing – review and editing

## Author ORCIDs

J Christopher Rounds  http://orcid.org/0000-0001-5287-8454
Ayan Banerjee  http://orcid.org/0000-0001-8350-851X
Milo B Fasken  http://orcid.org/0000-0003-4317-7909
Anita Corbett  http://orcid.org/0000-0002-0461-6895
Ken Moberg  http://orcid.org/0000-0002-9820-5543

## Decision letter and Author response

Decision letter https://doi.org/10.7554/eLife.64904.sa1
Author response https://doi.org/10.7554/eLife.64904.sa2

# Additional files

## Supplementary files

• Supplementary file 1. RNA abundance in *Nab2^ex3* vs. *control* female and male adult heads. This file contains a list of all annotated RNAs and their corresponding expression level across all samples in female and male adult heads (i.e., raw counts) and their fold-change in *Nab2* null heads relative to *control* in descending rank order of significance (p-value, column H) as determined by DESeq2 analysis.

• Supplementary file 2. Splicing changes in female and male *Nab2* null heads relative to control

heads. This file contains a list of RNAs and corresponding changes in splice patterns in *Nab2* null vs. *control* detected by MISO. A5SS, alternative 5′ splice site; A3SS, alternative 3′ splice site; RI, retained intron; SE, skipped exon.

- Supplementary file 3. DEXSeq called differential exon usage in *Nab2* null heads. This file contains a list of RNAs with differential exon usage in *Nab2* null female or male heads detected by DEXSeq in descending rank order of significance (p-value, column P).
- Transparent reporting form

### Data availability

Sequencing data have been deposited in GEO under accession code GSE162531.

The following dataset was generated:

| Author(s) | Year | Dataset title | Dataset URL | Database and Identifier |
|---|---|---|---|---|
| Jalloh B, Rounds JC, Corbett AH, Moberg KH | 2020 | The Nab2 RNA binding protein promotes sex-specific splicing of Sex lethal in *Drosophila* neuronal tissue | https://www.ncbi.nlm.nih.gov/geo/query/acc.cgi?acc=GSE162531 | NCBI Gene Expression Omnibus, GSE162531 |

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
