## [Editor Report]

This study provides important new insight into the function of Nab2 protein in regulating Sxl pre-mRNA splicing, a key developmental switch gene in *Drosophila*. By implicating Nab2 in the modulation RNA methylation on the Sxl transcript, the results illuminate the emerging role of methylation in the regulation of splicing. This proposed role for Nab2 in modulating m6A modification has implications for other systems.

---

## [Decision Letter]

**Decision letter after peer review:**

Thank you for submitting your article "The Nab2 RNA binding protein promotes sex-specific splicing of Sex lethal in *Drosophila* neuronal tissue" for consideration by *eLife*. Your article has been reviewed by 3 peer reviewers, and the evaluation has been overseen by Douglas Black as Reviewing Editor and K VijayRaghavan as the Senior Editor. The reviewers have opted to remain anonymous.

All of the reviewers found the manuscript to be of significant interest. The implication of Nab2 in Sxl splicing regulation adds a new facet to this important developmental pathway, and the proposed role Nab2 in modulating m6A modification has implications for other systems. However, the data supporting this role of m6a and the model for what Nab2 is doing are limited. Additional work is needed to support these conclusions.

Essential Revisions:

1. The support for the model would be greatly strengthened by forcing hypermethylation of Sxl pre-mRNA by an alternative method, independent of Nab2, and showing that indeed Sxl exon 3 skipping is affected in females. If feasible, the authors should assess genetic interactions of Nab2 with other genes in the m6A pathway. Making the Nab2/Mettl3 double mutant appears to have required some effort since they are closely linked. More description in the methods is needed as how this was made (how many lines were screened etc.). Have they tested interactions with YTHDC, whose locus is much further from Nab2? Can they test other components of the writer complex? The single genetic interaction with Mettl3 is interesting but not very deep support for their model of Nab2 function. The authors should discuss whether Nab2 protein interacts with components of the Mettl3 complex or with Virilizer.

2. The anti-m6A antibody pull-down of Sxl pre-mRNA is interesting, but why are m6A levels not more reduced in the Mettl3 mutant in Figure 6B and C? Given the Act5C and USP data it would appear that m6A levels are generally increased in Nab2 mutants. This needs better quantification. m6A antibodies often show limited specificity for modified over unmodified RNA or can generate high backgrounds from ribosomal or other RNA species (PMID: 32620324). Have these antibodies been assessed for the specificity of the pull-down? A chromatography approach (see Haussmann et al.) or some other method is needed to distinguish m6A in mRNA from that in rRNA and to more accurately assess the global m6A levels. Figure 6C should also include Nab2ex3::Mettl3null samples.

3. The Sxl splicing defects in the Nab2/Mettl3 double mutant need clearer analysis. The qPCR shown in Figure 5D/E is not sufficient. The Nab2 splicing defect is distinct from that seen in Mettl2 mutants. Is use of the cryptic 5' splice site in intron 3 reduced in the Nab2/Mettl3 double mutant compared to Nab2?

4. The authors use MISO to analyze splicing patterns in their RNA-seq data. While reliable, MISO is somewhat out of date and will only find splicing events that are pre-annotated in the MISO splice junction libraries. Similarly, measurement of exon reads in DEXSeq is an insensitive means of identifying splicing changes. It is surprising that alteration of so many splicing-related factors (32 in females and 75 in males) lead to such limited splicing changes in the brain (only 48 events in females and 50 in males). It is likely that many splicing changes were not identified. The identified alternative splicing events show a predominance of alternative 3' splice sites, which appears skewed from the common distribution of the different splicing patterns. Do they think Nab2 most commonly affects alternative 3' splice sites? Alternatively, is the number of events too small to yield an accurate distribution, or is there a bias in their detection? The authors might consider software that can identify unannotated splicing events (eg. PMID: 25480548, PMID: 29229983, PMID: 30104386).

5. It seems unlikely that the various neurological phenotypes observed in both Nab2 and Mettl3 mutants are mediated solely by Sxl mis-regulation. These phenotypes could be derived from misregulation of other Nab2 and/or m6A target genes. The levels of SXL protein should be shown in the brains of the mutant flies. Does removal of Sxl in mushroom bodies of females lead to projection defects? The possible relationship of the neurological phenotypes to splicing and dosage compensation defects should be more clearly discussed.

6. SXL not only regulates the translation of msl-2 mRNA, but also the splicing of a 5' UTR facultative intron that is retained in females. Although the analysis of splicing changes in Nab2ex3 flies does not retrieve this alternative splicing event, it may have been missed because of the small size of this intron. In fact, the IGV reads in Figure S4 across the msl-2 5' UTR intron appear somewhat different in control and Nab2ex3 female flies, contrary to what is stated. RT-qPCR should be used to confirm there are no differences in the retention of this intron, as this would change the interpretation of the effects on msl-2 and its consequences for female viability. Also msl-2 protein levels should be compared in control versus Nab2ex3 female brains. Please also describe the msl-2kmA allele, which is not an allele commonly used in msl-2 studies. Have the authors tested alternative msl-2 alleles (e.g. msl-2227/Df(2L)Exel7016 used in Haussmann et al., Nature 2016)?

---

## [Author Response]

Essential Revisions:1. The support for the model would be greatly strengthened by forcing hypermethylation of Sxl pre-mRNA by an alternative method, independent of Nab2, and showing that indeed Sxl exon 3 skipping is affected in females.

This was an excellent suggestion that we followed. As requested, we have added new data showing that overexpressing a *UAS-Mettl3* transgene encoding the catalytic subunit of the m^6^A transferase in neurons (*elav-Gal4*) of *wildtype* females is sufficient to drive (1) retention of exon 3 and (2) use of the cryptic 5’ splice site that produces the exon 3-intron3^503^-exon 4 RNA that we also detect in *Nab2* mutant female heads. These data are in an updated Figure 6 (panel H) and new Supplemental Figure 10.

If feasible, the authors should assess genetic interactions of Nab2 with other genes in the m6A pathway.

This was also a very helpful suggestion. As requested, we have added new data showing that *Nab2* mutant phenotypes in females are dominantly suppressed by alleles of the m^6^A transferase subunits *virilizer* and the WTAP homolog *f(l)2d*. This result parallels the effect of *Mettl3* and strengthens the genetic data to support the link between Nab2 and Mettl3. These data are presented in updated Figure 5 (panels F-G).

Making the Nab2/Mettl3 double mutant appears to have required some effort since they are closely linked. More description in the methods is needed as how this was made (how many lines were screened etc.).

As requested, we have now included data showing PCR genotyping of two different isolates of the *Nab2^ex3^,Mettl3^null^* double mutant chromosome (new Supplemental Figure 6) and we have added detail to the Methods section on the number of recombinant lines that were screened, which was indeed a significant number.

Have they tested interactions with YTHDC, whose locus is much further from Nab2? Can they test other components of the writer complex? The single genetic interaction with Mettl3 is interesting but not very deep support for their model of Nab2 function.

This was also a very helpful suggestion. We speculated in our original manuscript that excess m^6^A in *Nab2* mutant female heads could lead to excess recruitment of an m^6^A ‘reader’ protein like Ythdc1 (the nuclear reader) which in turn disrupts post-transcriptional processing of Nab2 target RNAs. Here we have tested this prediction genetically, and found that heterozygosity for *Ythdc1* rescues developmental lethality of *Nab2* mutants. This data supports our model and has been added to an updated Figure 6 (panel H).

The authors should discuss whether Nab2 protein interacts with components of the Mettl3 complex or with Virilizer.

As noted in our original Discussion, we identified Virma, the mammalian version of Virilizer, as a protein that co-immunoprecipitates with the vertebrate Nab2 orthologue ZC3H14 from mouse brain lysates (Morris et al., 2017, *NAR*). As yet, we have not been able to detect a direct interaction between the proteins that would explain the ability of Nab2 to regulate m^6^A levels at specific sites in *Sxl*. However, a recent series of papers (He et al., 2023, *Science*, Uzonyi et al., 2023, *Molecular Cell*, Yang et al., 2022, *Nature Communications*, Obrdlik et al., 2019, *Cell Reports*, Singh et al., 2012, *Cell*) show that both fly Nab2 and human ZC3H14 associate with the exon junction complex (EJC), and that the EJC inhibit m^6^A deposition at specific sites in spliced mRNAs. Our views on how these data impact models of Nab2 function in the fly brain have been incorporated into the revised Discussion.

2. The anti-m6A antibody pull-down of Sxl pre-mRNA is interesting, but why are m6A levels not more reduced in the Mettl3 mutant in Figure 6B and C? m6A antibodies often show limited specificity for modified over unmodified RNA or can generate high backgrounds from ribosomal or other RNA species (PMID: 32620324). Have these antibodies been assessed for the specificity of the pull-down? A chromatography approach (see Haussmann et al.) or some other method is needed to distinguish m6A in mRNA from that in rRNA and to more accurately assess the global m6A levels.

We agree that a drop to near zero m^6^A on a given RNA is expected in *Mettl3* nulls, but assays of m^6^A levels/abundance using a variety of approaches routinely detect a less dramatic effect due to the presence of other m^6^A modifying enzymes. For example, the same SynapticSystems anti-m^6^A antibody detects ‘m6A’ sites in CLIP analysis of RNA harvested from *Mettl3^null^* animals that should lack all m^6^A (Kan et al., 2021). As noted by the reviewer, this effect may be due to cross-reactivity of this antibody to 6mA (DNA) and/or m6Am (RNA). More to the point, these limitations of the anti-m^6^A pulldown approach led us apply an independent biochemical technique to map m^6^A sites in the *Sxl* RNA. This approach is termed deamination adjacent to RNA modification targets (DART) and was developed by Kate Meyer (Duke Univ., now a co-author) to precisely map m^6^A sites in specific RNAs. Our new DART data provide an independent test of m^6^A levels and location in *Nab2^ex3^* heads, demonstrating that m6A modification at specific sites is increased in *Nab2^ex3^* heads. These new data are presented in updated Figure 6 (panels D-F) and corroborate the anti-m^6^A pulldown data.

Given the Act5C and USP data it would appear that m6A levels are generally increased in Nab2 mutants. This needs better quantification.

We agree that it would be informative to define the full list of RNAs that become hypermethylated in the absence of Nab2, and how ‘general’ or ‘restricted’ this list is. Notably, our new DART analysis of *Sxl* RNA (see Figure 6D-F) reveals that some Mettl3 sites are hypermethylated in *Nab2^ex3^* mutant heads while others are not, indicating that Nab2 does not inhibit m^6^A at all *Sxl* sites that are targets of Mettl3. To address the issue of where Nab2 regulates m^6^A across *all* RNAs, we are currently collaborating with the Meyer Lab at Duke to perform a DART-seq analysis the wildtype and *Nab2* mutant head transcriptomes. We regard this important project as a follow-up to the *Sxl* data and thus beyond the scope of the current manuscript.

Figure 6C should also include Nab2ex3::Mettl3null samples.

While transferring stocks from the previous authors to the new trainees who performed all the experiments for this revision, we determined that ‘*Nab2^ex3^,Mettl3^null^*’ double homozygotes used in our first submission were incorrectly genotyped and were actually *Nab2* homozygous and *Mettl3* heterozygous (i.e., *Nab2^ex3^/Nab2^ex3,^Mettl3^null^/+*). We subsequently rederived two *Nab2^ex3^,Mettl3^null^* balanced stocks (new Supplemental Figure 6) and reperformed all experiments in validated *Nab2^ex3^/Nab2^ex3,^Mettl3^null^/+* animals. These new genotypes behaved identically to the older *Nab2^ex3^/Nab2^ex3,^Mettl3^null^/+* animals in all relevant assays, with the exception of suppressing *Sxl* splicing defects and MB defects. As the *Nab2^ex3^,Mettl3^null^* chromosome is homozygous lethal, we were unable to test whether complete loss of Mettl3 suppresses MB or.

We have also used *elav-Gal/UAS-RNAi* to lower Mettl3 levels in *Nab2^ex3^* female neurons; this also suppresses *Nab2^ex3^* lethality and behavioral phenotypes, consistent with Nab2 and Mettl3 coregulating neuronal RNAs. These data have been added to an updated Figure 5 (panels D-E).

3. The Sxl splicing defects in the Nab2/Mettl3 double mutant need clearer analysis. The qPCR shown in Figure 5D/E is not sufficient.

These panels have been removed and replaced by the new *Sxl* DART m^6^A mapping data (Figure 6D-F).

The Nab2 splicing defect is distinct from that seen in Mettl2 mutants.

The reviewer is correct that although *Nab2* and *Mettl3* loss both cause exon 3 inclusion, the 503nt intron 3 fragment that we detect in *Nab2^ex3^* nulls was not been reported in the prior three studies of *Sxl* splicing in *Mettl3* mutants (Lence et al., 2016 *Nature*; Haussmann et al., 2016, *Nature*; Kan et al., 2017, *Nat Comm*). However, our unpublished analysis of RNA-seq data deposited in the NCBI Sequence Read Archive by one of those groups (Kan et al., 2017) shows evidence that this 503nt intron 3 fragment is also retained in *Mettl3* mutant female heads. Moreover, our own data show that this fragment is also present in *wildtype* female heads but at much lower read depth than in *Nab2* mutants (red arrowheads in Figure below), leading us to speculate that it represents a short-lived *Sxl* splicing intermediate in females that accumulates in the absence of Nab2 or Mettl3. We have provided these IGV traces in Author response image 1 but consider these *Sxl* data to be secondary to our main conclusions regarding roles of Nab2.

**Author response image 1. sa2fig1:** 

Is use of the cryptic 5' splice site in intron 3 reduced in the Nab2/Mettl3 double mutant compared to Nab2?

Given the lethality of *Nab2^ex3^,Mettl3^null^* double mutants, we performed this assay on *Nab2^ex3^/Nab2^ex3^,Mettl3^null^/+* female heads and did not observe a statistically significant rescue of the E2-E3 or E3-E4 splicing defects (new Supplemental Figure 10). We speculate that loss of a single copy of Mettl3 may not be sufficient to rescue the splicing defect observed.

4. The authors use MISO to analyze splicing patterns in their RNA-seq data. While reliable, MISO is somewhat out of date and will only find splicing events that are pre-annotated in the MISO splice junction libraries. Similarly, measurement of exon reads in DEXSeq is an insensitive means of identifying splicing changes. It is surprising that alteration of so many splicing-related factors (32 in females and 75 in males) lead to such limited splicing changes in the brain (only 48 events in females and 50 in males). It is likely that many splicing changes were not identified. The identified alternative splicing events show a predominance of alternative 3' splice sites, which appears skewed from the common distribution of the different splicing patterns. Do they think Nab2 most commonly affects alternative 3' splice sites? Alternatively, is the number of events too small to yield an accurate distribution, or is there a bias in their detection? The authors might consider software that can identify unannotated splicing events (eg. PMID: 25480548, PMID: 29229983, PMID: 30104386).

We thank the Reviewer for suggesting approaches to deepen our bioinformatic analysis. We have used a modified pipeline to reanalyze the RNA-seq data and have added the output data, which do contain more predicted splicing defects, in a new Supplemental Material file. This more sensitive approach still identifies *Sxl* RNA as the most statistically significant ‘hit’ among all the Nab2-regulated transcripts, which was our rationale to focus on *Sxl* regulation in this study. We would also like to point out that the effect of Nab2 loss on splicing of specific RNAs is very robust, and that all of the approximately 150 top ranked hits easily pass the ‘eye test’ without the aid of software packages (e.g., see *CG13124* and *I_h_ channel* in Supplemental Figure 3) designed to detect more subtle, but nonetheless significant, effects. This is a main reason why we chose to focus on the top ranked eye-test ‘hit’ *Sxl* for our mechanistic studies.

More broadly, we are currently collaborating with the Meyer Lab at Duke to perform a DART-seq analysis the wildtype and *Nab2* mutant head transcriptomes, which will allow us to parse splicing defects that map to sites of excess m^6^A and those that may be Nab2-dependent but m^6^A independent.

5. It seems unlikely that the various neurological phenotypes observed in both Nab2 and Mettl3 mutants are mediated solely by Sxl mis-regulation. These phenotypes could be derived from misregulation of other Nab2 and/or m6A target genes.

We absolutely agree and have emphasized this point in the revised Discussion. We cite links we have found between Nab2 and other cellular pathways that do not involve the Sxl protein (e.g., planar cell polarity in Lee et al., 2020 and Corgiat et al., 2022) and highlight the oxidoreductase Wwox, which accumulates in the brains of *Nab2* mutant flies and *ZC3H14* mutant mice (Rha et al., 2017, Corgiat et al., 2020) and whose RNA contains a Nab2-regulated 3’UTR intron (this study) and a candidate m^6^A site (Kan et al., 2021).

The levels of SXL protein should be shown in the brains of the mutant flies.

To more systematically analyze the impact on Nab2 loss on Sxl RNA *and* protein expression, we have included a quantitative RT-qPCR analysis of the *Sxl* transcript using primers that detect either correctly (exon 2-exon 4) or incorrectly (exon 2-exon 3) spliced *Sxl*, as well as analyzing Sxl protein in female heads. These analyses show that *Nab2^ex3^* decreases exon 2-exon 4 splicing while increasing exon 2-exon 3 splicing, and that this switch in RNA structure is accompanied by a statistically significant, 2-fold drop in Sxl protein detected by western blot in multiple biological replicates. These new data are presented in updated Figure 3 (panels C-E).

Does removal of Sxl in mushroom bodies of females lead to projection defects? The possible relationship of the neurological phenotypes to splicing and dosage compensation defects should be more clearly discussed.

This comment seems to be based on one panel of data comparing MB morphology in *Nab2^ex3^/Nab2^ex3^* heads to *Nab2^ex3^/Nab2^ex3^,Mettl3^null^/+* heads which, as noted above, has been removed. However, spurred in part by the Reviewer’s query about Sxl, we tested its role in MB Kenyon cells using a MB-Gal4 in combination with a *UAS-Sxl* RNAi and were quite surprised to see defects in female but not male MBs. If validated, this result warrants further study beyond the scope of this study as it implies different mechanisms of MB development in males and females.

As requested, we have added text in the Discussion on the potential role of DCC dysregulation on axon projection defects in the female brain.

6. SXL not only regulates the translation of msl-2 mRNA, but also the splicing of a 5' UTR facultative intron that is retained in females. Although the analysis of splicing changes in Nab2ex3 flies does not retrieve this alternative splicing event, it may have been missed because of the small size of this intron. In fact, the IGV reads in Figure S4 across the msl-2 5' UTR intron appear somewhat different in control and Nab2ex3 female flies, contrary to what is stated. RT-qPCR should be used to confirm there are no differences in the retention of this intron, as this would change the interpretation of the effects on msl-2 and its consequences for female viability. Also msl-2 protein levels should be compared in control versus Nab2ex3 female brains.

We agree that our IGV data (Supplemental Figure 4) show evidence that this 5’UTR intron is spliced out in a fraction of transcripts in *wt* and *Nab2^ex3^* males, but we consider any effect in females to be very subtle. To follow this observation further, we tested Msl-2 protein levels in the relevant male and female genotypes by anti-Msl2 western blotting of whole head lysates. This experiment did not detect obvious differences in Msl-2 levels in wt vs. *Nab2* mutant male or female heads (data included in Author response image 2). Notably, the genetic suppression of *Nab2^ex3^* female phenotypes by *msl-2* alleles (Figures 4 and S5) does suggest that Msl-2 protein is produced and active in female brains, but this could occur in a subset of brain cells that is undetectable by western blot of whole heads.

Please also describe the msl-2kmA allele, which is not an allele commonly used in msl-2 studies. Have the authors tested alternative msl-2 alleles (e.g. msl-2227/Df(2L)Exel7016 used in Haussmann et al., Nature 2016)?

As requested, we have added a description of the *kmA* allele in the text and repeated the experiments using the suggested *msl-2^227^/Exel7016* allele combination. These new data support the *msl-2^kmA^* result and have been added to updated Figure 5 (panel A).